# Functionally heterogeneous human satellite cells identified by single cell RNA sequencing

Emilie Barruet[1†], Steven M Garcia[1†], Katharine Striedinger[1], Jake Wu[1], Solomon Lee[1], Lauren Byrnes[2], Alvin Wong[1], Sun Xuefeng[3], Stanley Tamaki[1], Andrew S Brack[3], Jason H Pomerantz[1]*

[1]Departments of Surgery and Orofacial Sciences, Division of Plastic and Reconstructive Surgery, Program in Craniofacial Biology, Eli and Edythe Broad Center of Regeneration Medicine, University of California, San Francisco, San Francisco, United States; [2]University of California San Francisco, San Francisco, United States; [3]Department of Orthopedic Surgery, Eli and Edythe Broad Center of Regeneration Medicine, University of California, San Francisco, San Francisco, United States

**Abstract** Although heterogeneity is recognized within the murine satellite cell pool, a comprehensive understanding of distinct subpopulations and their functional relevance in human satellite cells is lacking. We used a combination of single cell RNA sequencing and flow cytometry to identify, distinguish, and physically separate novel subpopulations of human PAX7+ satellite cells (Hu-MuSCs) from normal muscles. We found that, although relatively homogeneous compared to activated satellite cells and committed progenitors, the Hu-MuSC pool contains clusters of transcriptionally distinct cells with consistency across human individuals. New surface marker combinations were enriched in transcriptional subclusters, including a subpopulation of Hu-MuSCs marked by CXCR4/CD29/CD56/CAV1 (CAV1+). In vitro, CAV1+ Hu-MuSCs are morphologically distinct, and characterized by resistance to activation compared to CAV1- Hu-MuSCs. In vivo, CAV1 + Hu-MuSCs demonstrated increased engraftment after transplantation. Our findings provide a comprehensive transcriptional view of normal Hu-MuSCs and describe new heterogeneity, enabling separation of functionally distinct human satellite cell subpopulations.

*For correspondence:
jason.pomerantz@ucsf.edu

†These authors contributed equally to this work

Competing interests: The authors declare that no competing interests exist.

## Introduction

In mammalian skeletal muscle, tissue resident muscle stem cells, called satellite cells are characterized by location between the sarcolemma and the basal lamina, and by expression of the transcription factor PAX7. In mouse and human muscle, surface markers have been used to isolate and purify satellite cells, as a homogeneous population and distinct from more differentiated myogenic progenitors and differentiated muscle cells (*Collins et al., 2005*; *Kuang et al., 2007*; *Mauro, 1961*; *Montarras et al., 2005*; *Sacco et al., 2008*; *Sherwood et al., 2004*; *Alexander et al., 2016*; *Charville et al., 2015*; *Uezumi et al., 2016*; *Xu et al., 2015*). Several studies have investigated quiescent, activated and proposed 'satellite stem' cells within the mouse satellite cell pool, suggesting the existence of functionally distinct satellite cells (*Chakkalakal et al., 2014*; *Chakkalakal et al., 2012*). Increasing information from studies of mouse satellite cells suggests that the satellite cell pool is heterogeneous – only subsets of satellite cells are stem cells capable of self-renewal, whereas others commit to proliferation and muscle differentiation in a hierarchy that may include quiescent stem cells with self-renewal capacity, activated progenitors that commit to proliferation and differentiation, and senescent stem cells, as well as other potential intermediates (*Tierney and Sacco, 2016*;

*Der Vartanian et al., 2019*; *Scaramozza et al., 2019*). However, a comprehensive understanding of satellite cell heterogeneity in humans is lacking. Moreover, whether heterogeneous satellite cells exist simultaneously, and whether their proportions change in various physiological states or aging, is unknown. The absence of a prospective method to physically separate transcriptionally distinct, naturally occurring, satellite cells, impedes investigation and perpetuates a nebulous understanding of the satellite cell-state positions within the myogenic hierarchy, and how they are maintained under basal homeostatic conditions.

Although transcriptome profiles have been published for murine (*Aguilar et al., 2016*; *Alonso-Martin et al., 2016 Liu et al., 2013b*; *Machado et al., 2017*; *Pala et al., 2018*; *Pallafacchina et al., 2010*; *Ryall et al., 2015*; *Sousa-Victor et al., 2014*; *van Velthoven et al., 2017*; *Dell'Orso et al., 2019*) and human (*Charville et al., 2015*; *Rubenstein et al., 2020*) pooled satellite cells, transcriptional profiling of individual satellite cells has not yet been widely utilized to study muscle stem and early progenitor cells. Single cell RNA sequencing of 21 mouse satellite cells suggested transcriptional heterogeneity (*Cho and Doles, 2017*), and single-cell mass cytometry of murine SC shed light on transition states from quiescence through activation and differentiation (*Porpiglia et al., 2017*). Advances in droplet-based RNA sequencing have made it possible to analyze thousands of single cells with high fidelity and have been used to study non-muscle stem cells (*Aizarani et al., 2019*; *Fan et al., 2019*; *Macosko et al., 2015*; *Satija et al., 2015*; *Wagner et al., 2016*). Recently, single cell transcriptional profiling of mouse satellite cells and progenitors confirmed the core cell types of the myogenic differentiation pathway: quiescent and activated satellite cell, primary myoblast and committed progenitor (*Dell'Orso et al., 2019*). In this study, we utilize single cell RNA sequencing to characterize human satellite cells isolated from resting muscle. We demonstrate previously unappreciated functional heterogeneity within the satellite cell pool of healthy uninjured muscle. Furthermore, we show that transcriptome information can be used to select representative surface markers for physical separation of subpopulations. This approach, led to identification of novel subpopulations of human satellite cells and downstream in vitro and in vivo experimentation that demonstrated functional heterogeneity.

## Results

### Single cell RNA-seq of human satellite cells reveals distinct subpopulations within normal muscle

We used our previously developed approaches to isolate human satellite cells from fresh human muscle biopsies (*Garcia et al., 2018*; *Garcia et al., 2017*; *Xu et al., 2015*). Recent studies using mouse (*Machado et al., 2017*; *Der Vartanian et al., 2019*; *Scaramozza et al., 2019*) and human (*Charville et al., 2015*; *Cho and Doles, 2017*) cells have provided evidence that satellite cells have heterogeneous patterns of gene expression. To discern satellite cell subpopulations in normal human muscle, we performed single cell RNA sequencing on highly purified CXCR4+/CD29+/CD56 + human satellite cells (*Garcia et al., 2018*) from eight vasti lateralis muscle of human subjects (20–83 years old) (*Figure 1a* and *Figure 1—figure supplement 1a*). A total of 68,108 cells were analyzed (See *Figure 1—figure supplement 1b,c* for QC data). Utilizing the Seurat single cell analysis package (*Macosko et al., 2015*; *Butler et al., 2018*) we found 17 distinct clusters of cells, as represented in 2D uniform manifold approximation and projection (UMAP) *Figure 1b*. While we found the distribution of each cluster to vary among different samples we didn't find unique clusters associated with age or sex (*Figure 1c*). We confirmed clusters 0–8, 10, 12 and 15 to contain the vast majority of the cells (91%) and to consist of satellite cells by virtue of the expression of *PAX7* and *MYF5*, while clusters 11, 13, 14, 16 and 9 contained small numbers of contaminating cells that were *PAX7*- and expressed either a mesenchymal/fibroblastic/smooth muscle and hematopoietic pattern, or a more differentiated muscle signature (*Figure 1d,e* and *Figure 1—figure supplement 2a*). Thus, 12 transcriptionally distinct satellite cell clusters were identified to be present across vastus lateralis biopsies.

Each cluster was found to have a unique transcriptomic fingerprint with heterogeneous gene expression, *Figure 1e*. Each cluster was characterized by the top differentially expressed genes, shown in *Figure 1e* and *Supplementary file 1*. Cluster 0 was characterized by upregulation of genes associated with the NOTCH pathway (*DLK1*; *Guruharsha et al., 2012*), G-protein signaling (*GNAS*;

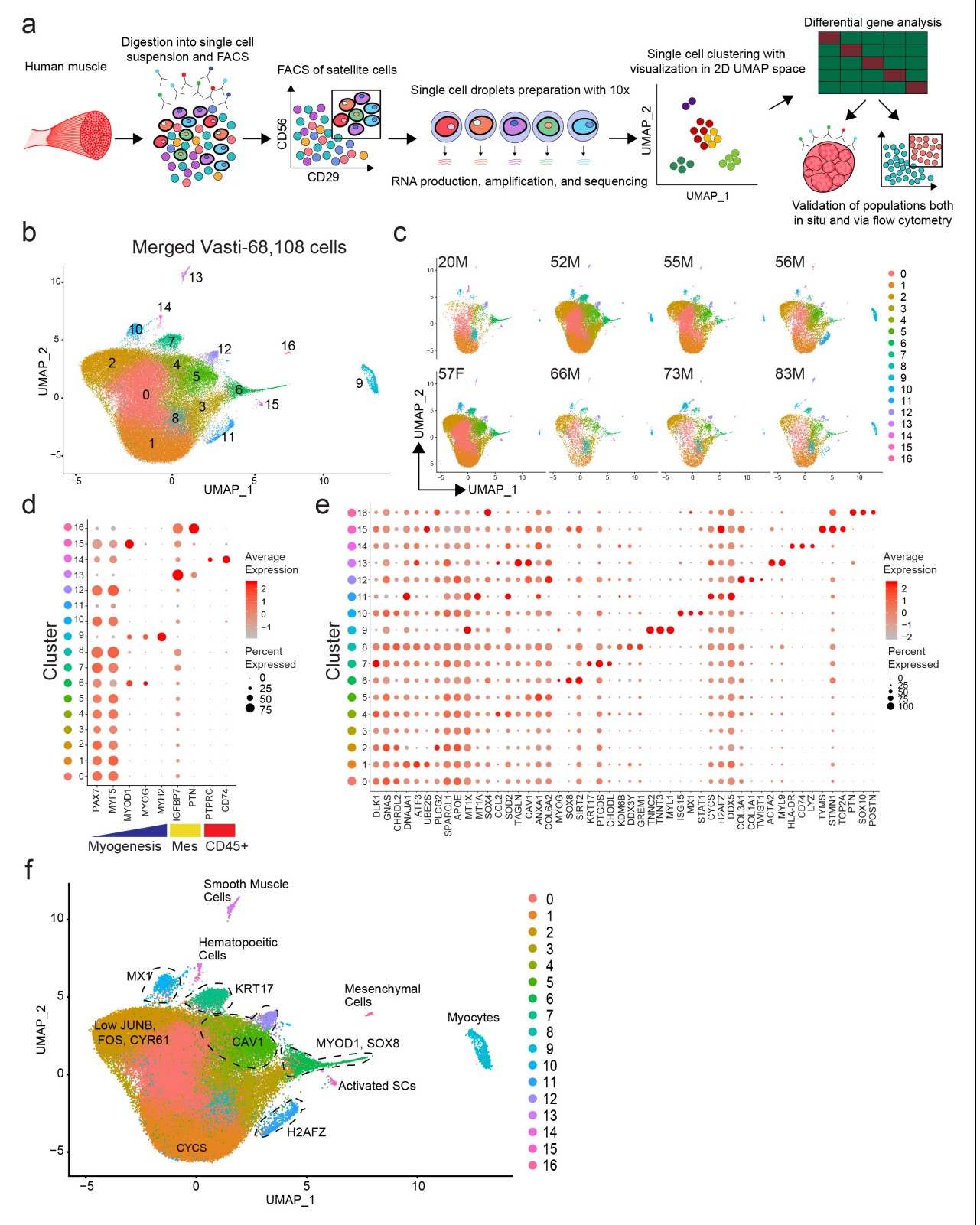

**Figure 1.** Single cell RNA sequencing reveals heterogeneity of the human satellite cell transcriptome. (a) Schematic diagram of the process of isolating human satellite cells from muscle biopsies, and then performing single cell RNA-seq to develop single cell transcriptomes leading to discovery of cell clusters, followed by in vivo validation. (b) UMAP plot of 68,108 cells isolated from eight vasti lateralis of male and female healthy subjects. Cells are clustered according to transcriptome similarity in 2D space. Each dot represents one cell, which are colored by cluster as identified by clustering

*Figure 1 continued on next page*

*Figure 1 continued*

analysis. (c) UMAP of each individual sample showing the distribution of cells in clusters for each sample. (d–f) Dot plots displaying expression of individual genes within each cluster. Each cluster is depicted on the y-axis and genes are labeled on the x-axis. Larger dot size represents more cells of that cluster expressing each gene; while color indicates the level of expression within those cells. (d) Dot plot displaying expression of genes associated with myogenesis, mesenchymal and hematopoietic markers in clusters 0–16. (e) Dot plot displaying the expression of top differentially expressed cluster markers in clusters 0–16. (f) 'Road-UMAP' with labeled cell types and key significantly regulated markers.

The online version of this article includes the following source data and figure supplement(s) for figure 1:

**Source data 1.** Single cell RNA sequencing reveals heterogeneity of the human satellite cell transcriptome.

**Figure supplement 1.** Flow cytometric sorting of CXCR4+/CD29+/CD56+ human satellite cells and quality control data from single cell RNA-sequencing experiments.

**Figure supplement 2.** PAX7/3 expression in vasti lateralis and transcriptome from a rectus femoris muscle of human satellite cells.

*Weinstein et al., 2004*) and satellite cell quiescence (*CHRDL2*; *Charville et al., 2015*). Cluster 1 up-regulated genes were associated with cellular stress response (*DNAJA1*; *Stark et al., 2014*, *ATF3*; *Hai et al., 1999*) and cellular development (*EGR1*; *Zhang et al., 2018*). Cluster 2 had high representation of genes involved in hormone signaling and inflammation (*APOE*; *Liu et al., 2013a*; *Tzioras et al., 2019*, *SPARCL1*; *Hurley et al., 2015*, *PLCG2*; *Yu et al., 2005*). Cluster 3 was found to have up-regulated genes associated with response to metal ions and oxidative stress (*MT1X*, *MT1A*; *Si and Lang, 2018*, *SOX4*; *Pan et al., 2017*). Cluster 4 was characterized by upregulation of genes associated with response to INF gamma (*CCL2*; *Schroder et al., 2004*), TNF signaling (*SOD2*; *Yi et al., 2017*, *TAGLN*; *Yang et al., 2009*). Cluster 5 was composed of cells expressing genes implicated in extra-cellular matrix (ECM), membrane receptor, focal adhesion (*COLs*, *CAV1*; *Yeh et al., 2017*, *ANXA1*; *Sheikh and Solito, 2018*) and down regulation of senescence. Cluster 6 consisted up-regulated genes associated with cellular development and mesenchymal cell development and differentiation (*SIX1*; *Wu et al., 2014*, *SOX8*; *Schmidt et al., 2003*) and myogenic commitment and differentiation (*MYOD1*, *MYOG*; *Dumont et al., 2015*). Based on the expression of *MYOD1*, cells in this cluster were considered activated and or progressing toward, differentiation. Cluster 7 was found to have cells with expression patterns of genes of positive regulation of cellular development, *IGF1* (*Mourkioti and Rosenthal, 2005*; *Schiaffino and Mammucari, 2011*), *KRT17* (*Karantza, 2011*) previously not described to be expressed by satellite cells, *PTGDS* (*Moniot et al., 2014*), and quiescence (*CHODL*; *Machado et al., 2017*). Cluster 8 up-regulated genes were associated with transcriptional repression (*TXNIP*; *Elgort et al., 2010*), and cellular stress response (*KDM6B*; *Mallaney et al., 2019*, *GREM1*; *Simeckova et al., 2019*). Cluster 10 contained cells expressing *MX1*, recently described to be present in a subpopulation of satellite cells (*Scaramozza et al., 2019*), and cell and genes associated with response to IFN gamma (*STAT1*; *Qing and Stark, 2004*, *ISG15*; *Deng et al., 2015*). Cluster 12 consisted of cells with expression patterns of genes of positive regulation of cell junction assembly (*CAV1*; *Song et al., 2007*, *THY1*; *Kumar et al., 2016*, *S100A10*; *Lee et al., 2004*), collagen catabolic processes, extracellular matrix assembly/disassembly (*TIMP1*, *MMP2*; *Arpino et al., 2015*), response to TGF-beta (*CAV1*; *Gvaramia et al., 2013*, *COLs*; *Biernacka et al., 2011*, *CLDN5*; *Shen et al., 2011*), blood vessel development (*GPC3*; *Ng et al., 2009*, *HSPG2*; *Martinez et al., 2018*, *CAV1*; *Yu et al., 2006*, *COLs*; *Marchand et al., 2018*, *THY1*; *Inoue et al., 2016*, *MMP2*; *Brooks et al., 1996*, *TIMP1*; *Arpino et al., 2015*, *B2M*; *Smith et al., 2015*). Cluster 15 consisted of a small number of cells expressing genes associated with DNA replication and cell cycle control (*TOP2A*, *BIRC5*, *MKI67*) suggesting satellite cell activation. We also evaluated expression of *PAX3*, which has been shown to be enriched in a subset of Pax7+ mouse satellite cells (*Der Vartanian et al., 2019*; *Scaramozza et al., 2019*). We found low detectable *PAX3* expression present all the myogenic clusters however cells expressing *PAX3* did not form a unique cluster (*Figure 1—figure supplement 2a*).

The differentiated muscle cluster (9), hematopoietic cluster (14) and fibrogenic/mesenchymal clusters (13,16) were confirmed to have expression profiles with up-regulated genes consistent with their identities respectively (*Figure 1e* and *Supplementary file 1*). Cluster 11 was made of cells expressing high levels of DNA damage expressing genes (*CYCS*; *Pal et al., 2010*, *H2AFZ*; *Flint et al., 2007*, *DDX5*; *Nicol et al., 2013*) and ribosomal genes suggesting low quality cells. We also analyzed human satellite cells from the rectus femoris (a different muscle from the quadriceps group) of an 84 year old male (*Figure 1—figure supplement 2b*). From this individual sample, a total of 4,575 cells

were analyzed (*Figure 1—figure supplement 1c*). Similar to the analysis of eight combined vasti, we confirmed clusters to consist of satellite cells by the expression of *PAX7* and *MYF5* (Clusters 0–3), while clusters 4 and 5 contained a small number of cells that were *PAX7*- and expressed either a mesenchymal/fibroblastic/endothelial pattern, or a more differentiated muscle signature, respectively (*Figure 1—figure supplement 2c*). Analogous cluster markers to the vasti samples were found in this rectus femoris sample (i.e. clusters expressing genes associated with NOTCH signaling (0), cellular stress (1), response to IFN gamma (cluster 2-*CCL2, MYC, ICAM1, IRF1*) (*Schroder et al., 2004*), ECM and cell adhesion (cluster 3-*CAV1, COLs, VCAM1*) (*Figure 1—figure supplement 2d* and *Supplementary file 2*). Taken together, these scRNAseq results show that satellite cells in human uninjured muscle, can be separated into transcriptionally distinct subpopulations (*Figure 1f*).

## Validation of cluster-specific markers and identification of satellite cell subpopulations in vivo

We selected highly expressed markers to validate the predicted human satellite cell subpopulations at the protein level, by immunofluorescence. We mapped five of the top differentially expressed genes (*CYCS, DLK1, ICAM1* and *VCAM1*) onto the 2D UMAP space, (*Figure 2a*). These genes are heterogeneously expressed across the eight vasti lateralis (*Figure 1e*), the rectus femoris sample, two pooled recti abdominis and two pooled pectoralis major samples (*Figure 2a* and *Figure 2—figure supplement 1a,b,c*). We then assessed proportions of cells with each marker detectable in human muscles by both immunofluorescence staining and flow cytometry (*Figure 2b* and *Figure 2—figure supplement 1d* and *Figure 2—figure supplement 2a–e*). Muscle sections from human tissue were stained with PAX7 and LAMININ to identify PAX7 expressing satellite cells. Sections were also stained with each cluster marker, as shown in *Figure 2b* and *Figure 2—figure supplement 2e*. With all five markers, we found PAX7 satellite cells in individual muscles that were either negative or positive for expression of each respective protein, confirming in vivo heterogeneity of sublaminar satellite cells. We also confirmed the detection of the surface markers ICAM1, DLK1 and VCAM1 on subsets of satellite cells isolated from human muscle using flow cytometry (*Figure 2—figure supplement 2a–e*). This utilization of flow cytometry for validation of surface markers demonstrates the feasibility of separating and isolating heterogeneous subpopulations of human satellite cells for downstream use and experimentation. The proportions of satellite cells expressing each cluster marker were quantified and compared among data from immunofluorescence staining, flow cytometry, and the number of expressing cells within the in silico data (*Figure 2c*). We discovered the proportion of PAX7+ satellite cells expressing DLK1, ICAM1, CYCS, and VCAM1 by immunofluorescence staining to be 35.0 ± 1.7%, 38.8 ± 1.9%, 49.7 ± 1.7%, and 10.6 ± 2.8%, respectively. In addition, we also found heterogeneous co-expression of those markers (DLK1+/ICAM1+, 28%; ICAM1+/VCAM1+, 21% and DLK1+/VCAM1+, 16%) in PAX7+ cells (*Figure 2—figure supplement 2d, e*). The portion of CXCR4+/CD29+/CD56+ cells expressing ICAM1, DLK1, and VCAM1 measured by flow cytometry was found to be 63.9 ± 13.8%, 37.0 ± 0.6%, and 13.5 ± 4.3%, respectively. Our results confirmed that differentially expressed genes (*DLK1, ICAM1, CYCS*, and *VCAM1*) identified via scRNAseq are expressed heterogeneously by human satellite cells in vivo.

## Quiescence and activation signature of human sorted satellite cells

We next evaluated the transcriptional signatures associated with stemness, quiescence and activation in the previously identified satellite cell clusters 0–8, 10, 12 and 15, as well as differentiated myogenic cells from cluster 9 (*Figure 3a*). We evaluated the expression of cell cycle markers previously described as part of the quiescent stem cell gene signature of satellite cells (*Cheung and Rando, 2013*). We found one cluster (15) composed of fully activated satellite cells (0.52%) with increased expression of G2/M/S phase markers such as *ANLN, BIRC5, CCNA2, CCNB1* or *CCNE2* while in the rest of the satellite cell clusters these were downregulated (*Figure 3b*). This was also confirmed by the cell cycle scoring vignette from Seurat and the increased RNA counts in cluster 15 (*Figure 3—figure supplement 1a,b*). Moreover, *Ki67* was only detected in cluster 15 providing additional evidence that the majority of satellite cells were not activated. We noticed an increase of the RNA count in cluster eight despite low levels of activation-associated cell cycle genes although this cluster also showed increased level expression of *MEF2C* compared to the rest of satellite cell clusters (*Figure 3c*). As expected, the cluster consisted of differentiated myocytes and showed

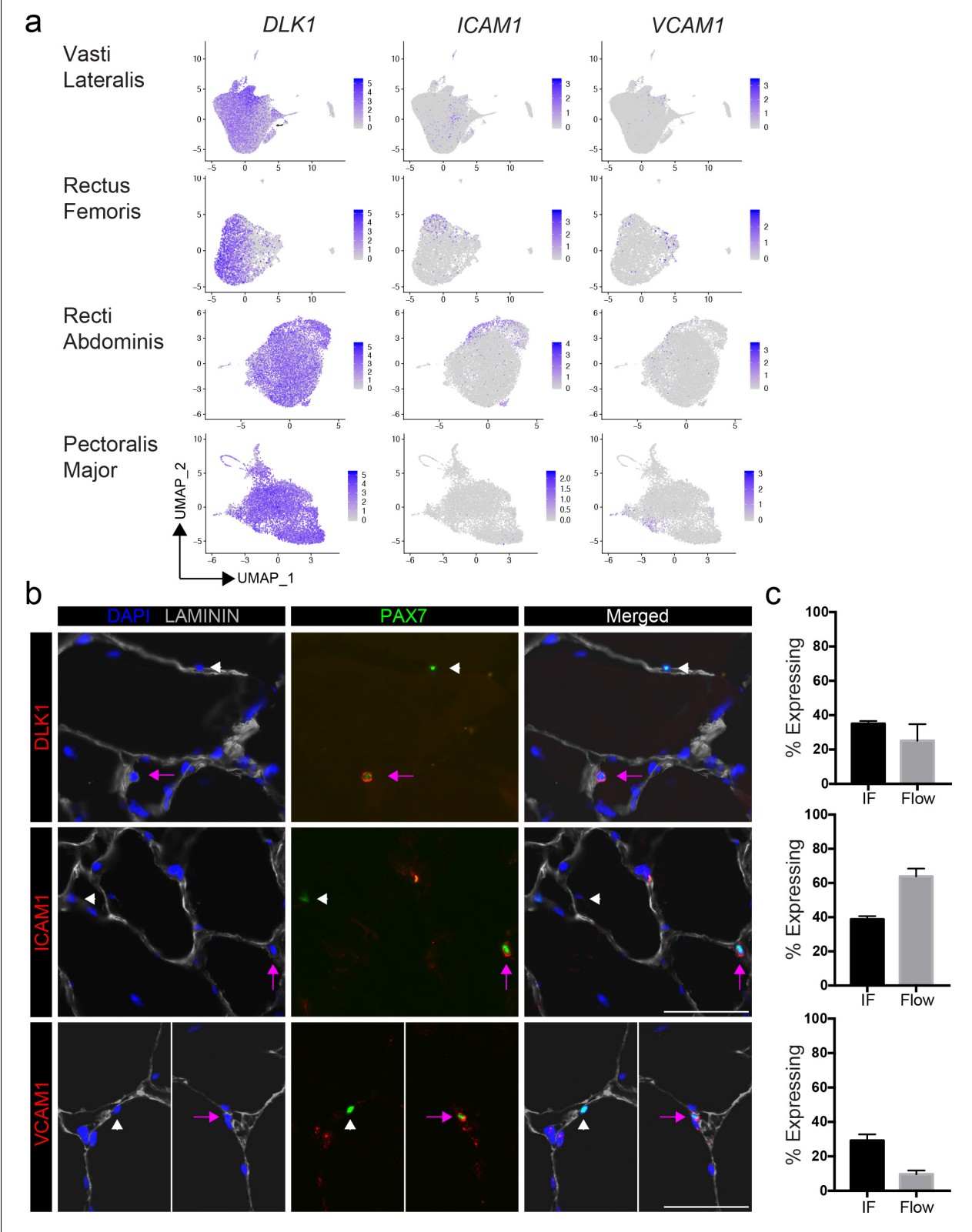

**Figure 2.** Validation of satellite cell clusters. (a) Feature plots displaying localized gene expression for the genes DLK1, ICAM1, and VCAM1 within the 2D UMAP space of the eight combined vasti, the rectus femoris, the two combined rectus abdominis and the two combined pectoralis majors as shown in *Figure 1b* and *Figure 1—figure supplement 2b*, *Figure 2—figure supplement 1a,b*. Each dot represents a single cell. Deeper purple coloration represents increased expression. (b) Immunofluorescence staining of human satellite cells for validation markers within sections of human muscle. Pink

*Figure 2 continued on next page*

*Figure 2 continued*

arrows denote human satellite cells that are positive for the marker (scale, 50 μm). White arrowheads mark satellite cells that are negative for expression of the marker. (n = 3, biological replicates). (c) Bar plot displaying the quantification of DLK1, ICAM1, and VCAM1 expression of human satellite cells with both immunofluorescence staining and flow cytometry. (n = 3, biological replicates). Data presented as mean ± SEM.

The online version of this article includes the following source data and figure supplement(s) for figure 2:

**Source data 1.** Validation of satellite cell clusters.
**Figure supplement 1.** Validation of satellite cell cluster marker CYCS across multiple human muscle type.
**Figure supplement 1—source data 1.** Validation of satellite cell cluster marker CYCS across multiple human muscle type.
**Figure supplement 2.** Heterogeneous expression of ICAM1, VCAM1 and DLK1 in human satellite cells.

characteristics of non-proliferative cells (e.g. low expression of G2/M/S phase markers (*Figure 3b*), low RNA count (*Figure 3—figure supplement 1b*).

We also evaluated previously used markers of stemness (*KLF4*, *MYC*) (*Takahashi et al., 2007*), quiescence (*NDRG2*, *DAG1* and *CHRNA1* (*Charville et al., 2015*), *SPRY1* (*Bigot et al., 2015*; *Chakkalakal et al., 2012*), *Shea et al., 2010*), the NOTCH pathway targets *HEY1* (*Jiang et al., 2014*), *EGR1* (*Min et al., 2008*), *HES1* (*Jiang et al., 2014*), *CD82* (*Alexander et al., 2016*), and activation (*MEF2C*; *Liu et al., 2014*) as shown in *Figure 3c*. Each satellite cell cluster demonstrated a mixed pattern of expression of those markers. We found differential expression among satellite cell clusters including *KLF4*, *MYC*, and *EGR1* (cluster 1); *HEY1* (cluster 12) and *HES1* (cluster 6, 9, 12); *CD82* (cluster 5, 12, 15)); *DAG1* (6, 15, 4) and *CHRNA1* (cluster 6, 12). Cluster six in which *MYOD1* and *MYOG* were differentially expressed also had *HES1*, *DAG1* and *CHRNA1* significantly upregulated. *SPRY1* was expressed in all of the satellite clusters at varying levels, while *MEF2C* was differentially upregulated in the myocyte cluster 9. In agreement with in silico analysis, immunofluorescence of human muscle tissue revealed that SPRY1 was detectable in a subset (~33%) of satellite cells in vivo (*Figure 3d*). We also found *SOX8*, a previously described satellite cell marker (*Schmidt et al., 2003*), to be associated with activation in human satellite cells (*Figure 3c*). Independent analysis of the rectus femoris resulted in similar findings (*Figure 3—figure supplement 1c*). Taken together, while we observed evidence of variable profiles consistent with quiescence, priming, or early activation, and notwithstanding a presumed effect of the purification and preparation process (i.e, cluster 6, 8, 15) the majority of satellite cells analyzed are best characterized by pre-activation, relatively quiescent states.

## Subpopulation relationships analyzed by pseudotime reflect transitions of early myogenic progression

In order to estimate the lineage relationships between the satellite cell clusters, we performed pseudotime analysis of our single cell data on all myogenic clusters utilizing the R package Monocle (*Qiu et al., 2017a*; *Qiu et al., 2017b*; *Trapnell et al., 2014*; *Figure 3e*). We found the cells ordered from proximal to distal in an arrangement compatible with quiescent satellite cells transitioning towards myogenic differentiation with several branching points (*Figure 3e*). Cells from cluster 9, which express genes of myogenic commitment and terminal differentiation were located distal to the satellite cell clusters at the end point of pseudotime while satellite cell clusters were distributed more proximally along sub-branches. The gene expression pattern across pseudotime suggests that two main populations of satellite cells diverge in accordance with expression of *PAX7*, *SPRY1*, *HEY1*, *DLK1* and *CAV1* (*Figure 3f*, dashed line). Analysis of the rectus femoris was confirmatory, with satellite cells diverging in multiple branches while the most distal branch consisted of differentiated myogenic cells. The branch represented by the solid line in *Figure 3—figure supplement 1d, e*) was driven by the expression of *PAX7*, *MYF5*, *HEY1* and *CAV1* (*Figure 3—figure supplement 1e*). Taken together, the pseudotime analysis shows that within resting human muscle, satellite cells can be ordered in various states of transition from stem-like cell to more differentiated progenitors. Transcription factors associated with more stem-like and quiescent, non-activated satellite cell states early in pseudotime (*PAX7*, *SPRY1*, *HEY1*) were associated with surface proteins DLK1 and CAV1, indicating a potential avenue for a physical sorting strategy.

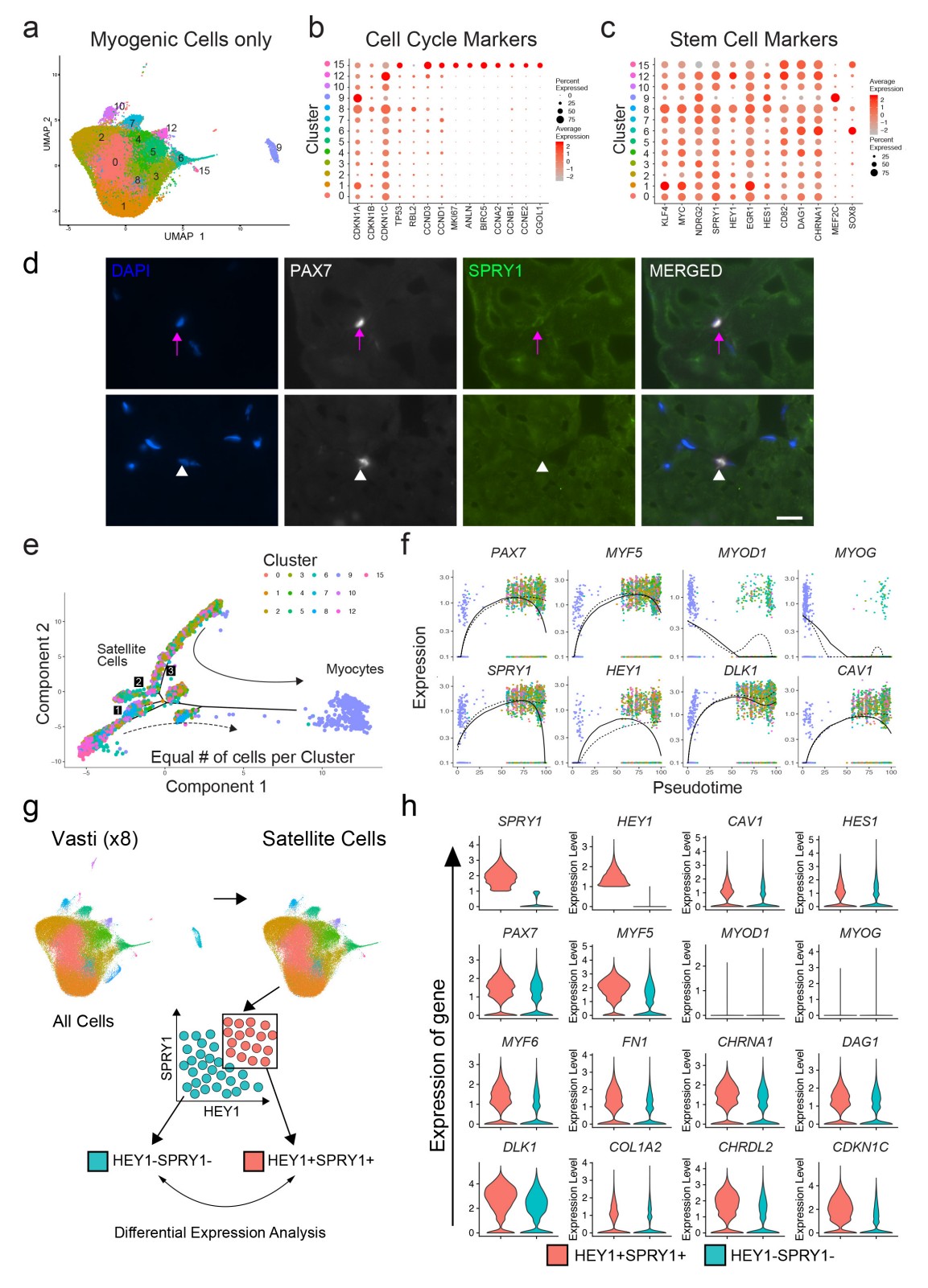

**Figure 3.** Quiescence molecular signature of human sorted satellite cells. (a) UMAP representing all myogenic cells used for downstream analyses. (b) Dot plot displaying the expression of cell cycle genes associated with satellite cell stemness, quiescence, and activation in the satellite cells for all myogenic clusters. (c) Dot plot displaying the expression of genes associated with satellite cell stemness, quiescence, and activation in the satellite cells. (d) Immunofluorescence staining of human satellite cells for SPRY1 within sections of human muscle. Pink arrows denote human satellite cells that

*Figure 3 continued on next page*

Figure 3 continued

are positive for the marker (scale, 20 μm). White arrowheads mark satellite cells that are negative for expression of SPRY1. (n = 3, biological replicates). (e) Pseudotime trajectories developed through Monocle analysis for the eight vasti. The object was downsampled and 500 cells per cluster were used. Plot cells were labeled by cluster as in (a). Arrows represent the direction of pseudotime. Branch points 1–3 are marked with numbers along the trajectories. (f) Gene plots displaying the expression of specific genes through branch point '3', as a function of pseudotime. (g) Schematic diagramming the in silico process for sorting satellite cells by expression of SPRY1 and HEY1 within the cells from the eight vasti. Satellite cells were first subclustered and separated by gene expression, followed by analysis to discover differentially expressed genes. (h) Violin plots displaying expressed genes in the SPRY1+HEY1+ and SPRY1-HEY1- experimental groups. Plots on the x-axis are colored by group. Specific gene expression is on the y-axis. The width of the violin plots depicts the larger probability density of cells expressing each gene particular gene at the indicated expression level.

The online version of this article includes the following source data and figure supplement(s) for figure 3:

**Source data 1.** Quiescence molecular signature of human sorted satellite cells.

**Figure supplement 1.** Quiescence state of human satellite cells and pseudotime trajectory.

## In silico sorting of quiescence signatures supports CAV1 as a marker and sorting target to separate functionally heterogeneous human satellite cell populations

We were interested in separating functionally distinct satellite cell subpopulations from normal adult muscle. We therefore decided to analyze the transcriptome data based on previously established markers of satellite cell quiescence: Sprouty1 (SPRY1) (*Shea et al., 2010*) and Hairy/enhancer-of-split related with YRPW motif protein 1 (HEY1) (*Fukada et al., 2011*). To do this, we performed an in silico grouping of the satellite cell populations in the adult sample dataset (clusters 0–8, 10, 12 and 15) excluding the rare contaminating and differentiated non-satellite cell populations. Cells were then sorted into two groups in silico by expression of SPRY1 and HEY1: SPRY1/HEY1 high expressing cells (SPRY1$^{hi}$/HEY1$^{hi}$) and SPRY1/HEY1 low/negative expressing cells (SPRY1$^{low/neg}$/HEY1$^{low/neg}$) (*Figure 3g*). The two groups were analyzed for differentially expressed genes with greater than 1.5-fold upregulation. In agreement with known MRF expression in quiescent and activated satellite cells, we found that both PAX7 and MYF5 were upregulated in the SPRY1$^{hi}$/HEY1$^{hi}$ group, while MYOD1, MYOG, and MYF6 were upregulated in the SPRY1$^{low/neg}$/HEY1$^{low/neg}$ group (*Figure 3h*). Indeed, several other genes associated with quiescence and activation were associated with the SPRY1$^{hi}$/HEY1$^{hi}$ group (Fibronectin 1, quiescence marker CHRNA1, DAG1, DLK1 and CHRDL2) and the SPRY1$^{low/neg}$/HEY1$^{low/neg}$ group (CYCS, EIF4A1, MTIX, PDK4, SOD2 and CYR61) respectively (*Supplementary file 3*). Moreover, caveolae scaffolding protein, Caveolin-1 (CAV1), which was associated with early pseudotime satellite cells, showed a larger probability density in the SPRY1$^{hi}$/HEY1$^{hi}$ cells. CAV1 has been shown previously to be expressed by mouse satellite cells (*Kann and Krauss, 2019*; *Gnocchi et al., 2009*) and expression is reported to be lost with activation (*Volonte et al., 2005*). This prompted us to perform in silico sorting of satellite cells by expression of CAV1: CAV1 high expressing cells (CAV1$^{high}$) vs CAV1 low expressing cells (CAV1$^{low}$) (*Figure 4a*). The average expression of quiescence marker SPRY1, HEY1, CD82, DAG1 and CHRNA1 was increased in CAV1$^{high}$ expressing satellite cells while the expression KLF4 and MYC were decreased (*Figure 4b*) further suggesting that CAV1 may be associated with satellite cells in a more quiescent state. The two groups were analyzed for differentially expressed genes. Differentially up- and down-regulated genes were subjected to Gene Ontology (GO) analysis and Pathway analysis (*Figure 4c,d* and *Supplementary file 4*). The GO analysis revealed that up-regulated genes were predominantly associated with extracellular matrix organization and cell-cell adhesion regulation while down-regulated genes were associated with myogenesis and stress responses. Pathway analysis showed that up-regulated genes in the CAV1$^{high}$ cells were associated with extracellular matrix, membrane receptors, VEGF signaling and focal adhesion. Down-regulated genes were associated with TGF-beta signaling. These gene expression analyses together suggest that CAV1 expression correlates with both quiescence and cell adhesion characteristics in human satellite cells.

## CAV1+ satellite cells are a phenotypically and functionally distinct human satellite cell subpopulation

CAV1 was detected in 55% of cells, however expression was significantly upregulated in clusters 4, 5, 12 and 15 across the eight vasti lateralis and rectus femoris, rectus abdominis and pectoralis major

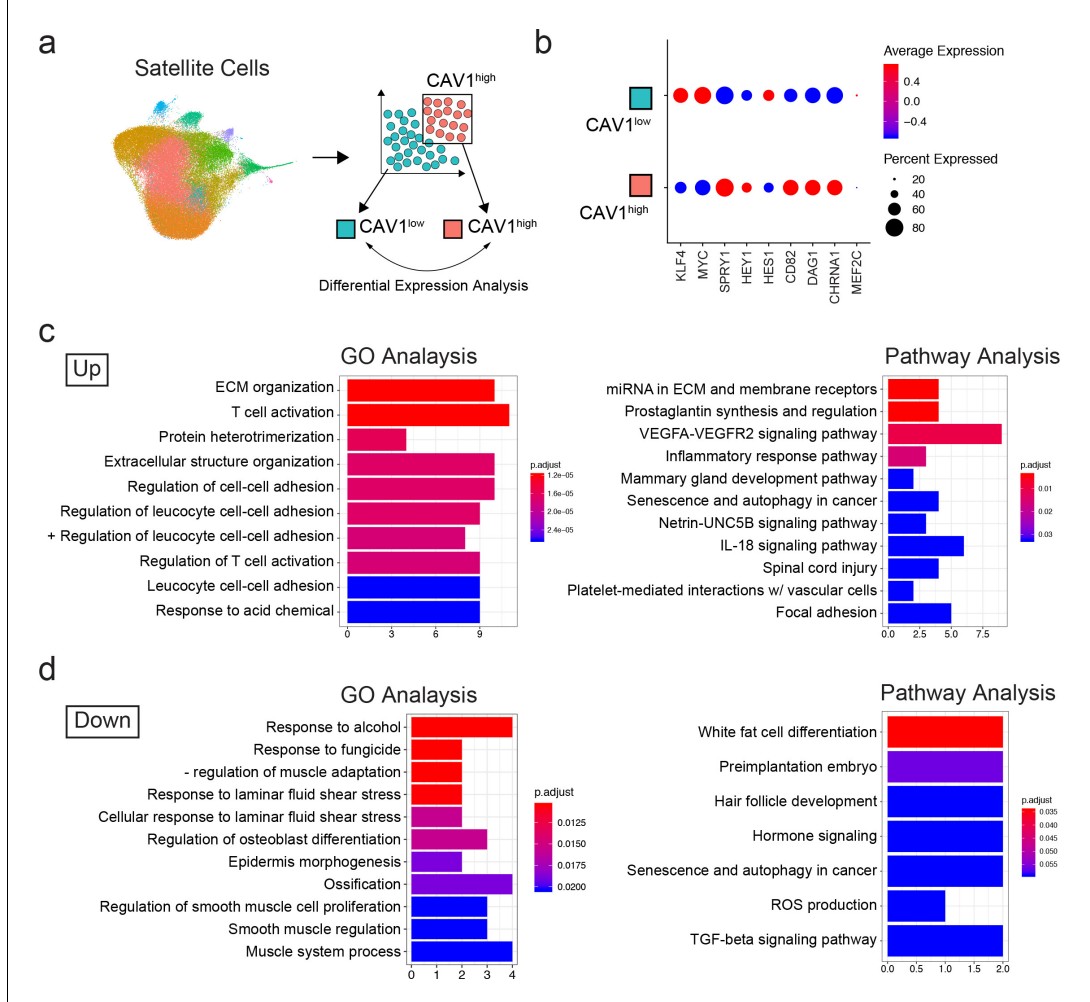

**Figure 4.** Transcriptomic analysis of *CAV1* high expressing cells . (a) Schematic diagramming the in silico process for sorting satellite cells by expression of *CAV1*. Satellite cells were first subclustered and separated by gene expression, followed by analysis to discover differentially expressed genes. (b) Dotplot displaying the average expression and percentage of cells expressing quiescence markers. (c–d) Gene ontology and Pathway analyses of differentially up- and down-regulated genes in *CAV1* high expressing satellite cells.

samples (*Figure 5a*). CAV1 protein expression was also heterogeneous when assayed in muscle sections and by flow cytometry of satellite cells (*Figure 5b and c*). We detected CAV1 expression on $10.65 \pm 3.6\%$ (n = 19) of satellite cells and $0.7 \pm 0.3\%$ of the whole cell population by flow cytometry (*Figure 5c*). The less frequent detection of CAV1 by flow cytometry of live cells reflects either the various intracellular localization possibilities of CAV1 (*Boscher and Nabi, 2012*), which are detected by immunofluorescence of sections, whereas flow cytometry will only detect CAV1 present on the cell surface (*Wu and Terrian, 2002*). Alternatively, CAV1 protein may be degraded rapidly upon SC isolation from the niche.

To physically separate CAV1 human satellite cells, CXCR4+/CD29+/CD56+ cells were sorted based on CAV1 expression (*Figure 5d*), marking human satellite cells with differential surface CAV1 expression (CAV1+ and CAV-). (*Figure 5f*) Upon back-gating analysis of the CAV1+ and CAV1- populations (*Figure 5e*) we found overlap in regard to cell size, cell granularity, and CXCR4, CD29 and CD56 expression, indicating that separation of these two populations is not attributable to other commonly assessed satellite cell characteristics. CAV1+ sorted cells expressed significantly higher transcriptional levels of *CAV1* compared to CAV1- (*Figure 5f*) confirming that flow cytometry captures CAV1$^{high}$ expressing satellite cells identified by scRNA sequencing. Moreover, both CAV1+ and CAV1- sorted satellite cells expressed PAX7 by immunofluorescence and qPCR (*Figure 5—*

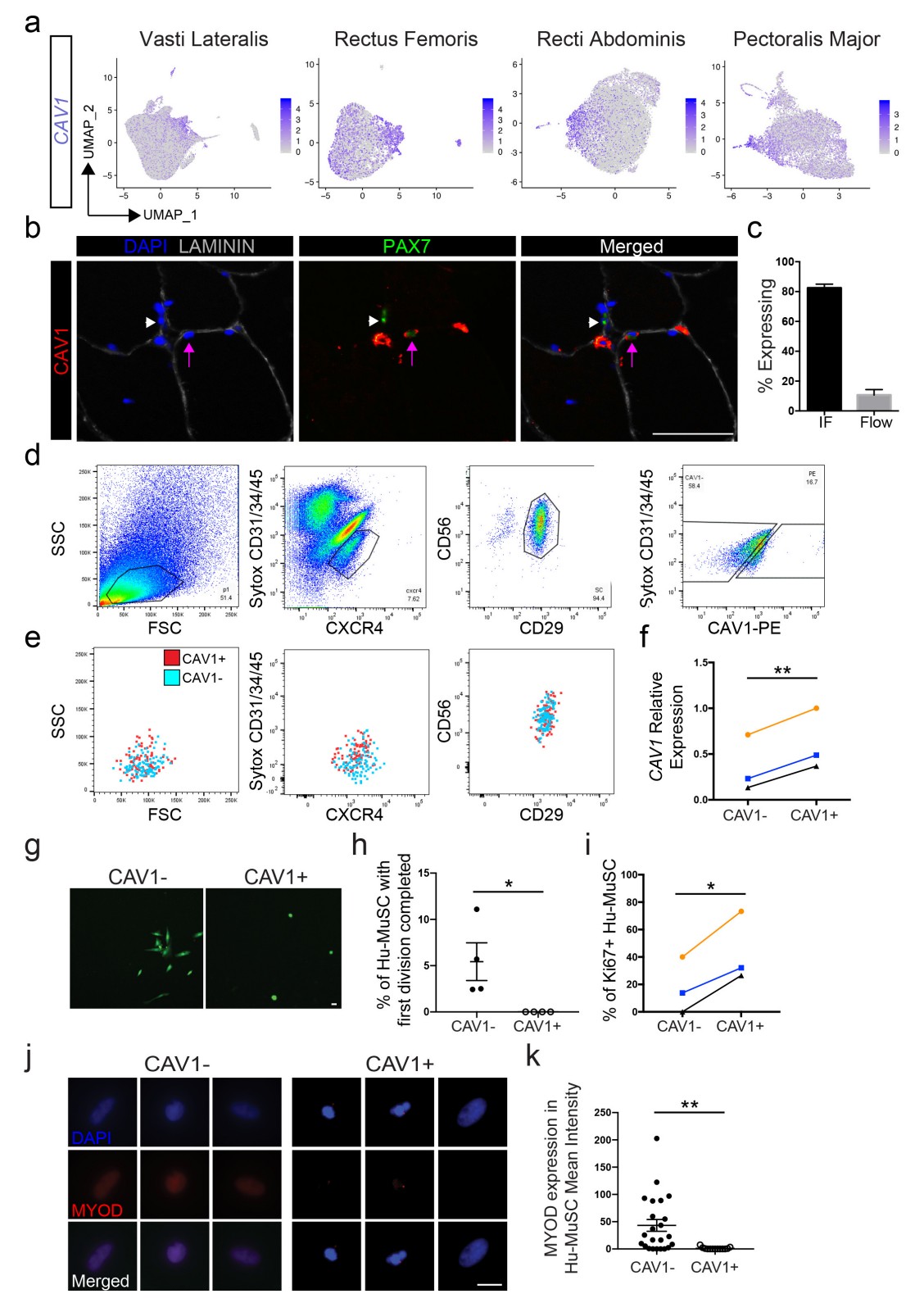

**Figure 5.** CAV1 high expressing human satellite cell phenotypes. (a) Feature plots displaying localized *CAV1* gene expression within the 2D UMAP space as shown in *Figure 1b*. Each dot represents a single cell. Deeper purple coloration represents increased expression. (b) Immunofluorescence staining of human satellite cells for CAV1 within sections of human muscle. Lavender arrows denote human satellite cells that are positive for the marker (scale, 50 μm). White arrowheads mark satellite cells that are negative for expression of the marker. (n = 3, biological replicates). (c) Bar plot displaying

*Figure 5 continued on next page*

Figure 5 continued

the quantification CAV1 expression of PAX7+ human satellite cells with both immunofluorescence staining and flow cytometry. (n ≥ 3, biological replicates). Data presented as mean ± SEM. (**d**) Representative flow cytometry profiles from the isolation of CXCR4/CD29/CD56 human satellite cells based on expression of CAV1. (**e**) Representative back-gating of CAV1+ (red) and CAV1- (blue) cells, demonstrating overlap of profiles within prior gates. (n = 3, biological replicates). (**f**). *CAV1* gene expression in sorted CAV1- and CAV1+[l] satellite cells (n = 3, biological replicates) *p<0.05. (**g**) Morphology of sorted CAV1- and CAV1+ satellite cells stained with CellTracker Green 4.6 days after isolation (scale, 20μm). (**h**) Timelapse analysis to assess time to first division. Quantification of the percentage of cells dividing in the first 6 days after isolation (n = 4, biological replicates, *p<0.05, mean ± SEM). (**i**) Percentage of cells expressing Ki67 3 days after isolation (n = 3, biological replicates, *p<0.05). (**j**) Immunofluorescence staining of MYOD 3 days after isolation (scale, 10 μm). (**k**) Quantification of MYOD expression at day 3 in vitro (n = 4, biological replicates, **p<0.01, mean ± SEM). Comparisons of CAV1+ and CAV1- satellite cells are from the same donor in each individual experiment.

The online version of this article includes the following source data and figure supplement(s) for figure 5:

**Source data 1.** CAV1 high expressing human satellite cell phenotypes.
**Figure supplement 1.** PAX7 analysis of CAV1- and CAV1+[l] sorted satellite cells.

figure supplement 1a,b), ruling out the possibility that CAV1- cells are non-satellite cell contaminants. This possibility is also contrary to the transcriptome data, which show that a large proportion of satellite cells are CAV1- or CAV1[low].

The ability to separate CAV1[high] expressing cells permitted us to test phenotypic and functional characteristics. CAV1+ and CAV1- satellite cells were sorted and cultured in growth-promoting conditions. The morphological phenotype differed significantly, with CAV1+ cells adopting a round shape compared to the more spindle and elongated morphology of CAV1- satellite cells (**Figure 5g**). We next used time lapse video microscopy to evaluate time to first division (**Kuang et al., 2007**; **Marti et al., 2013**; **Siegel et al., 2009**). Three separate experiments using satellite cells from four unique human muscles were performed by comparing CAV1+ and CAV1- satellite cells from the same muscle. Cells were sorted and then placed into culture and live stained with CellTracker Green for live visualization for 6 days. While none of the CAV1+ Hu-MuSC divided during this time, 5.4% (range 2.5–11%) of CAV1- Hu-MuSC completed their first cellular division in the first six days in vitro. The average time for CAV1 negative Hu-MuSC to complete their first division was 4.4 days (range 2.8–5.7 days). The time to first division was significantly longer in CAV1+ cells compared to CAV1- (**Figure 5h**). In support of this finding of slower division, we found that Ki67 expression is expressed in fewer CAV1+ Hu-MuSCs at day 3 of culture (**Figure 5i**). Finally, CAV1+ and CAV1- satellite cells were cultured in growth media for 3 days and MYOD expression was evaluated. CAV1+ Hu-MuSC expressed significantly less MYOD than their CAV- counterparts (**Figure 5j, k**). These in vitro assays are consistent with the previously discussed transcriptome data and indicate that CAV1+ cells differ from other human satellite cells in canonical assays of satellite cell activation and proliferation, collectively characterized by resistance to activation.

## Engraftment capacity after transplantation resides within the CAV1+ satellite cell subpopulation

The capacity to separate a subpopulation of satellite cells based on CAV1 expression enabled us to assay in vivo functional heterogeneity. From single muscle samples, 500 CAV1- and CAV1+ satellite cells were transplanted into the tibialis anterior (TA) of pre-irradiated NOD scid gamma (NSG) mice (**Garcia et al., 2018**; **Xu et al., 2015**). We (**Garcia et al., 2018**; **Xu et al., 2015**) and others have (**Arpke and Kyba, 2016**; **Gayraud-Morel et al., 2012**) previously demonstrated that the use of small number of cells are effective for transplantation studies. Mice were sacrificed 5 weeks later and evaluated for engraftment and myogenic differentiation using human specific DYSTROPHIN antibody (**Figure 6a**). The engraftment of human muscle was quantified by counting the maximum number of human specific DYSTROPHIN fibers per cross-section in each experimental group. We found that transplantation with CAV1+ human satellite cells led to robust engraftment in contrast to transplantation with equal numbers of CAV1- cells (**Figure 6b**). The CAV1+ transplants resulted in 4-fold higher engraftment over the CAV1- group (69.3 ± 18.4* vs 14.4 ± 6.3* human fibers respectively (p=0.013)), and the efficiency of human fibers corresponded to roughly 1 fiber per seven satellite cells transplanted. Repopulation of the satellite cell niche by human PAX7 cells was also significantly increased in the CAV1+ transplants when evaluated by counting human-derived PAX7 sublaminar cells on cross sections shown in **Figure 6c**, and quantified in **Figure 6d**. Finally, using the re-isolation

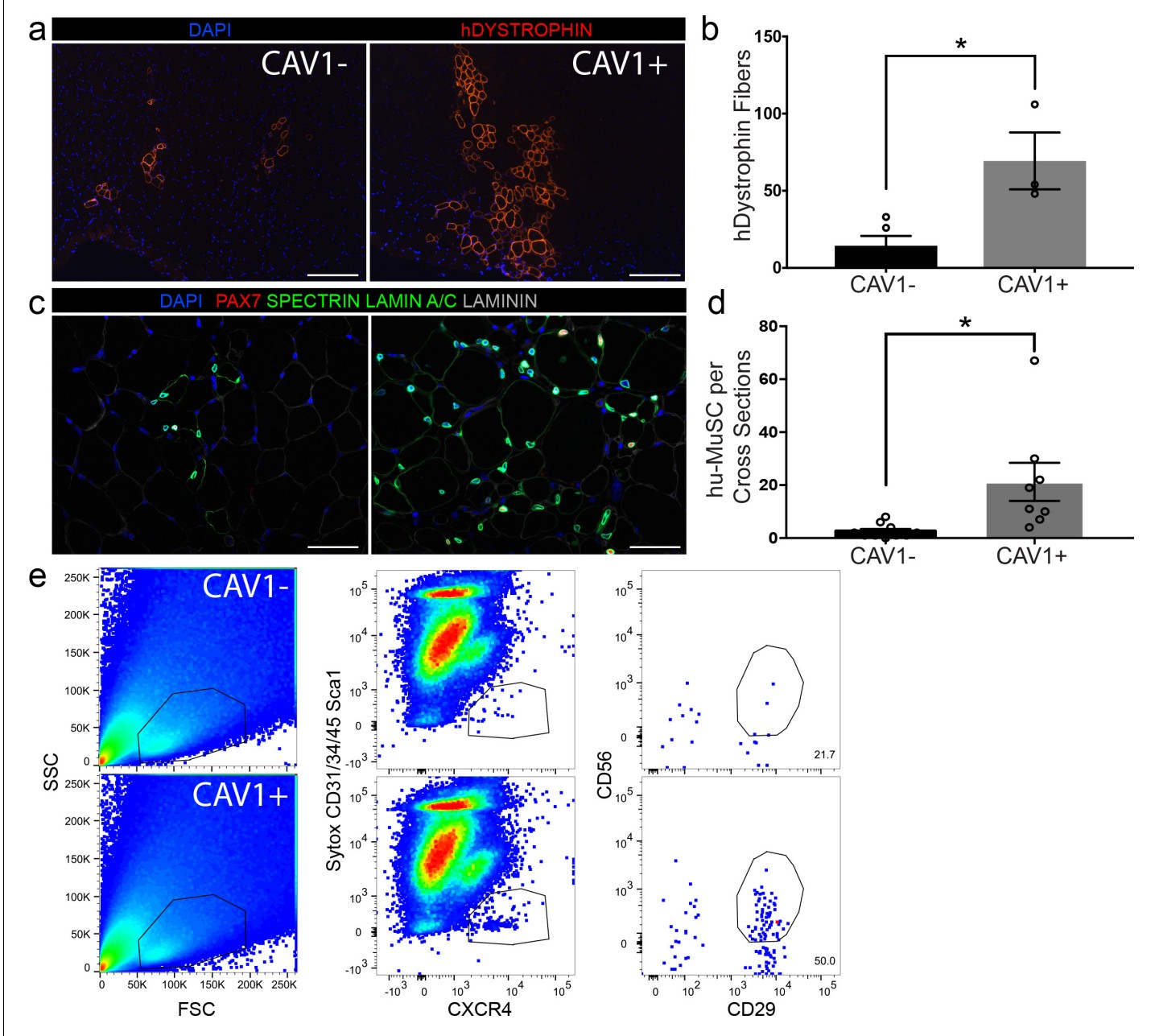

**Figure 6.** CAV1 high expressing human satellite cells engraft robustly after transplantation in mice. (**a**) Representative immunofluorescence staining images for human specific DYSTROPHIN (scale, 200 μm) within NSG mouse muscle cross-sections in TAs transplanted with CAV1+ and CAV1- human satellite cells. 500 cells were transplanted in each NSG TA. (n = 3, using three separate donors.) (**b**) Bar graph depicting the number of DYSTROPHIN positive human fibers in mice transplanted with CAV1+ and CAV1- cells. (n = 3, biological replicates) *p<0.05. Data presented as mean ± SEM. (**c**) Representative immunofluorescence staining images for human PAX7, SPECTRIN, LAMIN A/C and LAMININ within NSG mouse muscle cross-sections in TAs transplanted with CAV1+ and CAV1- human satellite cells (scale, 100 μm). (n = 3, biological replicates). (**d**) Bar graph represents number of human PAX7+ cells in mice transplanted with both CAV1+ and CAV1- cells. (n = 3, biological replicates) *p<0.05. Data presented as mean ± SEM. (**e**) Human satellite cells were re-isolated by FACS from mice transplanted with CAV1+ and CAV1- satellite cells.

The online version of this article includes the following source data for figure 6:

**Source data 1.** CAV1 high expressing human satellite cells engraft robustly after transplantation in mice.

assay we previously developed (*Garcia et al., 2018*) we were able to re-purify human satellite cells from mice transplanted with CAV1+ but not CAV1- cells (*Figure 6e*). In conclusion, human CAV1+ satellite cells engraft, differentiate, repopulate the satellite cell niche and retain satellite cell phenotypes efficiently and to a greater extent than the more numerous CAV1- satellite cells.

## Discussion

Human satellite cells have typically been purified and studied in bulk which leads to ensemble averaging of molecular and functional attributes. To identify precise mechanisms that control human stem cell function across a population requires assays that deconvolve heterogeneity. The findings of this study identify populations of transcriptionally heterogeneous satellite cells within the satellite cell pool of normal resting adult human muscle. Analysis of over 68,000 single cells led to identification of similar subpopulations in vastus lateralis muscle biopsies from different individuals, and in other muscle types from distinct donors, strongly suggesting that distinct subpopulations exist normally and in relatively stable states. Moreover, the unbiased analysis of numerous biological replicates indicates that similar subpopulations exist in common in different individuals. Single satellite cell data from separate individuals revealed remarkable homogeneity of the transcriptome compared to other transcriptomic analyses that readily distinguish disparate cell types, and compared to contaminating cells, activated satellite cells or committed myogenic progenitors, the latter differing markedly from the satellite cell population. This finding along with the pseudotime analysis, implies that skeletal muscle, while a low turnover tissue, contains transcriptionally diverse cells within the resident stem cell pool. Transcriptionally distinct satellite cell subpopulations were discernible and readily validated by protein expression in subsequent biopsies. The identification and validation of surface markers enriched in transcriptional subpopulations enables physical separation.

This report provides new insight into naturally occurring heterogeneity of human satellite cells. Although in vivo validation of expression of several genes supports the fidelity of the approach, once manipulated as in the case here by enzymatic digestion and flow cytometry, transcriptional signatures presumably change to some extent in accordance with previously published observations (*Machado et al., 2017*; *van Velthoven et al., 2017*). However, it is remarkable how different the majority of satellite cell signatures remain from those of activated satellite cells present in our samples. Thus the stimulation of transcriptional changes by isolation and sorting does not approach that of culture activated satellite cells or fully activated cells in vivo. Therefore, it can be concluded that isolation of satellite cells by the commonly used approach of enzymatic digestion and flow cytometry yields cells more resembling their natural states than what is seen after injury or even brief culture. This conclusion is supported by the validation assessing protein expression in muscle sections of biopsies. Regarding the satellite cell transcriptional subpopulations described here, there are two equally plausible conclusions with respect to satellite cell identities. It is possible that distinct subpopulations retain their identities throughout adult life as differentiated subtypes of satellite cells, and it is equally plausible that transcriptional subpopulations represent transient satellite cell states that transition across a continuum. While it is not currently feasible to distinguish these two possibilities using destructive assays at fixed timepoints, either model is consistent with our identification of relatively stable transcriptional subpopulations with distinct phenotypes and function.

Whereas an unbiased approach identified transcriptional clusters, a complementary strategy utilizing in silico sorting facilitated parsing of the relatively homogenous population into putative functionally distinct groups using surface markers. Differential expression analysis of cells sorted for quiescence genes, *SPRY1* and *HEY1*, led to identification of several other associated genes including *CAV1*. Our finding that human satellite cells are heterogeneous in CAV1 expression is in contrast to findings in mice which demonstrated that mouse satellite cells are uniformly CAV1 positive (*Ono et al., 2009*). This discrepancy could represent species differences in satellite cell marker expression and could also be influenced by differing subcellular localization or level of expression. CAV1 has been shown to play a role in processes that affect stem cell populations (*Baker and Tuan, 2013*), and warrants further investigation of its role in satellite cell physiological states and in supporting engraftment after transplantation. Transcriptional signatures of CAV1 cells are suggestive of a greater degree of quiescence, and this is strongly supported by in vitro data showing lower Ki67 expression, lower MYOD expression, and prolonged time to first division. Xenotransplantation with small numbers of CXCR4/CD29/CD56/CAV1+ human satellite cells resulted in robust engraftment

and residence of quiescent satellite cells in NSG mice. The levels of engraftment we observed with 500 cells are similar to those in prior experiments using thousands to a hundred thousand cells (*Charville et al., 2015*; *Garcia et al., 2018*; *Uezumi et al., 2016*; *Xu et al., 2015*). Whereas engraftment in vivo may relate to canonical stem cell properties such as quiescence, our data suggest that it may also relate to satellite cell adhesion properties. Indeed, it is notable that GO and pathway analysis of CAV1+ satellite cells showed multiple prominent representations of adhesion related pathways. Future studies will determine the properties of CAV1+ satellite cells that are responsible for engraftment potential.

In summary, this report provides a comprehensive view of baseline satellite cell transcriptional activity. Although satellite cell purification as well as other methods of preparing cells for transcriptome analysis is expected to cause some divergence from natural states, associated changes are significantly more modest than those seen in activated satellite cells or committed progenitors. Therefore, as we demonstrated and validated using the particular example of CAV1, strategic selection of surface markers informed by single cell transcriptome analysis is an effective approach to discern and investigate naturally occurring human satellite cell subpopulations.

# Materials and methods

## Key resources table

| Reagent type (species) or resource | Designation | Source or reference | Identifiers | Additional information |
|---|---|---|---|---|
| NSG mice | NOD.Cg-Prkdcscid Il2rgtm1Wjl/SzJ | https://www.jax.org/strain/005557 | 005557 | 8–12 week-old |
| Sequenced-based reagent | Human RT-PCR Primers | Applied Biosystems Taqman Assays | B-actin Hs01060665_g1 | |
| Sequenced-based reagent | Human RT-PCR Primers | Applied Biosystems Taqman Assays | CAV1 Hs00971716_m1 | |
| Sequenced-based reagent | Human RT-PCR Primers | Applied Biosystems Taqman Assays | PAX7 Hs00242962_m1 | |
| Antibody | Mouse monoclonal anti-Human DYSTROPHIN | DSHB | MANDYS104(7F7) | IF(1:10) |
| Antibody | Mouse monoclonal anti-Human/Mouse PAX7 | DSHB | PAX7 | IF(1:10) |
| Antibody | Rabbit polyclonal anti-LAMININ | Sigma-Aldrich | L9393 | IF(1:250) |
| Antibody | Mouse monoclonal anti-Human SPECTRIN | Leica Microsystems | NCL-SPEC1 | IF(1:100) |
| Antibody | Mouse monoclonal anti-Human LAMIN A/C | Vector Laboratories | VP-L550 | IF(1:100) |
| Antibody | Mouse monoclonal anti-Human/Mouse MYOD | BD Pharmigen | 554130 | IF(1:100) |
| Antibody | Rabbit polyclonal anti-CAV1 | abcam | ab2910 | IF(1:500) |
| Antibody | Mouse monoclonal anti-CAV1 (7C8) | Santa Cruz Biotechnology | sc-53564 | IF(1:50) |
| Antibody | Mouse monoclonal anti-DLK1 (MM0514-9D8) | abcam | ab89908 | IF(1:50) |
| Antibody | Mouse monoclonal anti-ICAM1 (G-5) | Santa Cruz Biotechnology | sc-8439 | IF(1:50) |
| Antibody | Rabbit monoclonal anti-VCAM1 (EPR5047) | abcam | ab134047 | IF(1:75) |

*Continued on next page*

*Continued*

| Reagent type (species) or resource | Designation | Source or reference | Identifiers | Additional information |
|---|---|---|---|---|
| Antibody | Mouse monoclonal anti-Human CYCS | LifeSpan Biosciences | LS-B6577 | IF(1:100) |
| Antibody | Mouse monoclonal anti-Human Ki67 | BD Pharmigen | 556003 | IF(1:100) |
| Antibody | Rabbit polyclonal anti-Human SPROUTY1 | abcam | ab111523 | IF(1:50) |
| Antibody | Mouse monoclonal anti-Human CD31 (Beads) | Miltenyi Biotec | 130-091-935 | FACS |
| Antibody | Mouse monoclonal anti-Human CD45 (Beads) | Miltenyi Biotec | 130-045-801 | FACS |
| Antibody | Mouse monoclonal anti-Human CD31 AF450 (WM-59) | Ebioscience | 48-0319-42 | FACS |
| Antibody | Mouse monoclonal anti-Human CD34 eFluor450 (4H11) | Ebioscience | 48-0349-42 | FACS |
| Antibody | Mouse monoclonal anti-Human CD45 AF450 (30-F11) | Ebioscience | 48-0451-82 | FACS |
| Antibody | Mouse monoclonal anti-Human CD29 FITC (TS2/16) | Ebioscience | 11-0299-41 | FACS |
| Antibody | Recombinant human anti-CD56 APC-vio-770 (REA196) | Miltenyi Biotec | 130-114-548 | FACS |
| Antibody | Mouse monoclonal anti-Human CXCR4 PE (12G5) | Ebioscience | 12-9999-41 | FACS |
| Antibody | Mouse monoclonal anti-Human CXCR4 APC (12G5) | Ebioscience | 17-9999-42 | FACS |
| Antibody | Rabbit monoclonal anti-Human CAV1 PE (EPR15554) | abcam | ab212007 | FACS |
| Antibody | Mouse monoclonal anti-Human ICAM1 PE (15.2) | abcam | ab210195 | FACS |
| Antibody | Mouse monoclonal anti-Human VCAM1 PE (STA) | Ebioscience | 12-1069-42 | FACS |
| Antibody | Mouse monoclonal anti-Human DLK1 PE (211309) | R and Dsystems | MAB1144 | FACS |
| Antibody | FcR block | Miltenyi Biotec | 130-059-901 | |
| Software, algorithm | GraphPad Prism | GraphPad Prism (https://graphpad.com) | | |
| Software, algorithm | ImageJ | ImageJ (http://imagej.nih.gov/ij/) | | |
| Software, algorithm | Seurat (3.1.2) | https://satijalab.org/seurat/ | | |
| Software, algorithm | Monocle (2.12.0) | http://cole-trapnell-lab.github.io/monocle-release/ | | |

*Continued on next page*

*Continued*

| Reagent type (species) or resource | Designation | Source or reference | Identifiers | Additional information |
|---|---|---|---|---|
| Software, algorithm | cellranger | https://support.10xgenomics.com/single-cell-gene-expression/software/pipelines/latest/feature-bc | | |
| Software, algorithm | FlowJo | https://www.flowjo.com | | |
| Cell Line (*Homo sapiens*) | HEK293 | ATCC Cat# PTA-4488, | RRID:CVCL_0045 | |

## Human muscle procurement

This study was conducted under the approval of the Institutional Review Board at The University of California San Francisco (UCSF). Biopsies were obtained from individuals undergoing surgery at UCSF. Written informed consent was obtained from all subjects. All types of muscle used for each experiment are listed in *Supplementary file 5*.

## Animal care and transplantation studies

All mice were bred and housed in a pathogen-free facility at UCSF. All procedures were approved and performed in accordance with the UCSF Institutional Animal Care and Use Committee. All experiments were unblinded and performed in 8–12 week-old NSG. Mice were randomized to all experimental groups by sex and littermates and were pretreated with 18 gamma (Gy) on the day before transplantation. Hu-MuSCs were injected along with 50 µl 0.5% bupivacaine directly into the TA muscle of one leg as indicated (*Garcia et al., 2017*). Detailed information can be found in the Supplemental Experimental Procedures section.

## CXCR4+/CD29+/CD56+ Satellite Cell Sorting

Freshly harvested human muscle was either immediately digested or stored in DMEM with 30% FBS at 4°C overnight. Muscle was digested, erythrocytes were lysed, and hematopoietic and endothelial cells were depleted with magnetic column depletion (Miltenyi Biotech). Viable cells were depleted for CD31, CD34, and CD45 expressing cells. Cells that remained after depletion were sorted for CXCR4+/CD29+/CD56+ and collected for further experimentation (*Garcia et al., 2018*; *Garcia et al., 2017*).

## Single cell RNA sequencing and analysis

To capture individual cells, we utilized the Chromium Single Cell 3' Reagent Version one and Version 3 Kit from 10X Genomics (*Zheng et al., 2017*). For all samples (vasti, rectus femoris, recti abdominis and pectoralis major) 18,000–30,000 satellite cells isolated as in *Garcia et al. (2018)* were loaded onto one well of the 10X chip to produce Gel Bead-in-Emulsions (GEMs). GEMs underwent reverse transcription to barcode RNA before cleanup and cDNA amplification. Libraries were prepared with the Chromium Single Cell 3' Reagent Version 1and 3 Kit (see *Figure 1—figure supplement 1b,c*). Each sample was sequenced on 1 lane of the HiSeq2500 (Illumina) run in Rapid Run Mode with paired-end sequencing parameters or 1 lane of the NovaSeq 6000 S4. The estimated number of cells, mean reads per cell, median genes per cells, median UMI (Unique Molecular Identifier) counts per cells as well as other quality control information are summarized in *Figure 1—figure supplement 1b,c*. Gene-barcoded matrices were analyzed with the R package Seurat v3.1, (*Satija et al., 2015*; *R Development Core Team, 2014*; *Zheng et al., 2017*). Cells with fewer than 500 genes, greater than 6000 genes and genes expressed in fewer than 5 cells were not included in the downstream analyses. We also filtered cells that had more than 10% mitochondrial counts. In all samples UMI counts (or RNA counts) were normalized with NormalizeData using default settings. The Find-VariableFeatures function was used to determine subset of feature that exhibit high cell-to-cell variation in each dataset based on a variance stabilizing transformation ('vst'). We used the default setting returning 2000 feature per dataset. These were used for downstream analysis. In the case of the merged data analysis samples were combined utilizing the FindIntegrationAnchors function with

the 'dimensionality' set at 30. Then, we ran these 'anchors' to the IntegratData function for batch correction for all cells enabling them to be jointly analyzed. The resulting outputs were scaled and UMI counts and mitochondrial contamination regressed out with the ScaleData function. We didn't regress out heterogeneity associated with cell cycle stage since it is an important factor in determining the state of quiescence of our sorted human satellite cells. PCA was performed with RunPCA, and significant PCs determined based on the Scree plot utilizing the function PCElbowPlot. The resolution parameter in FindClusters was adjusted to 0.5. Clusters were visualized by UMAP with Seurat's RunUMAP function. Differentially expressed genes were determined with the FindAllMarkers function. We performed differential gene-expression utilizing Seurat v3's FindMarkers function with the Model-based Analysis of Single-Cell Transctiptomics (MAST) that treats cellular detection as a covariate to calculate adjusted p values for multiple comparisons. Lists of differentially expressed genes for individual analyses are provided in *Supplementary file 1* and *2*. The cell-cycle scoring vignette from Seurat v3 was used to calculate cell cycle phase scores for each cell based on its expression of G2/M and S phase markers. Cells that didn't express G2/M and S phase markers were scored as not cycling cells in G0/G1 phase. Scores were assigned using the CellCycleScoring function and visualized in a barplot for each cluster. The in silico FACS were done using the subset function for each gene of interest. The differential expression was performed as described earlier.

## Pseudotime ordering
We utilized Monocle 2.12.0 to order cells in pseudotime based on their transcriptomic similarity (*Qiu et al., 2017b*). Variable genes from Seurat analysis were used as input and clusters were projected onto the minimum spanning tree after ordering. For computing power purposes, the combined vasti object was downsampled to 500 cells per cluster. Gene expression patterns were plotted with plot_genes_branched_pseudotime, and plot_multiple_branches_pseudotime function.

## Immunofluorescence
Cells were plated immediately after sorting on Matrigel coated chamber slides. 3 hr after plating cells were stained for PAX7 (DSHB). CD56-CD29-CXCR4- cells were used as controls. MYOD and Ki67 protein expression were assessed 3 days after plating. Cryosection slides or sorted human satellite cells were fixed with 4% PFA at room temperature for 10 min, washed in PBST (Phosphate Buffered Saline Tween20 0.1%), permeabilized with 0.1%Triton-100X (Sigma-Aldrich) and then blocked with protein-free serum block (DAKO) or 2% goat serum and incubated at room temperature overnight with primary antibodies (Supplemental Experimental Procedures and Key Resources Table). After PBST wash the corresponding secondary antibodies were applied for 1 hr at room temperature. Sections were mounted with VECTASHIELD mounting medium with DAPI (Vector Laboratories) and all samples were examined using a Leica upright or DMi8 Leica microscope.

## Cell line
We used the HEK293 cell line, ATCC Cat# PTA-4488, as a control in *Figure 5—figure supplement 1*. The cell line is tested periodically for mycoplasma and is negative to date.

## RT-PCR and quantitative analysis
Tissues were collected in RLT buffer (Qiagen), total RNA was isolated using the RNAeasy isolation kit (Qiagen). RNA was transcribed into cDNA with High-Capacity cDNA Reverse Transcription kit (ThermoFisher Scientific). cDNA was then pre-amplified with GE PreAmp Master Mix (Fluidigm Inc). Real-time quantitative PCR was performed in triplicated with either Taqman Universal PCR Master Mix (Life Technologies) on either a Viia7 thermocycler (Life Technologies). Taqman primers are listed in the Key Resources Table. Beta actin was used for normalization as endogenous control.

## Time lapse microscopy
Hu-MuSCs were sorted and plated at a density of 500 to 1000 cells per well on a 48 glass well plate (Mattek) precoated with Matrigel (Corning). Hu-MuSCs were grown in Growth media: DMEM high glucose phenol free media, 20% FBS and 1% pen/strep (Gibco). The following day, satellite cells were incubated with 10 mM final concentration of CellTracker Green CMFDA dye (Thermofisher Scientific) in phenol free media for 45 min at 37°C to track cell division. After a media wash, fresh

growth media was added for subsequent time lapse experiments, cells were imaged using Zeiss Confocal Microscope. Images of Hu-MuSC were taken every 15 min for 6 days. Images and videos were analyzed using Zeiss Zen microscope software. Statistical analysis was done using GraphPad Prism.

### Statistical analysis

Normality of the data was checked utilizing the Shapiro-Wilk normality test in GraphPad Prism. Means between or across groups were compared using two-tailed t-tests for experiments involving two groups, or one-way ANOVA with post hoc Tukey multiple comparisons when comparisons were made across three or more groups to determine significance ($p < 0.05$) between test conditions and controls, and multiple groups. Multivariate regression was utilized as indicated for comparing satellite cell yield per gram controlling for age, gender, and muscle type. All human muscle samples collected over the past one year and processed within 12 hr after biopsy were used for data analyses in *Figure 1*. At least three mice per group were used for all transplantation experiments. At least three biological replicates (three different muscle source) for each experiment were performed unless otherwise noted, with exact *n* values listed in each figure legend. For CAV1+/- statistical analysis paired or unpaired t-tests were used. All error bars are depicted as s.e.m. p-values are indicated with asterisks (*$p < 0.05$, **$p < 0.01$, ***$p < 0.001$).

### Supplemental experimental procedures
#### Animal care and transplantation studies

All mice were bred and housed in a pathogen-free facility at UCSF. All procedures were approved and performed in accordance with the UCSF Institutional Animal Care and Use Committee. All experiments were unblinded and performed in 8–12 week-old NOD.Cg-Prkdcscid Il2rgtm1Wjl/SzJ (NSG) mice (The Jackson Laboratory). Mice were randomized to all experimental groups by sex and littermates and were pretreated with 18 gamma (Gy) on the day before transplantation. A 5 mm incision was made in the mouse skin overlying the TA muscle. We used the multiple injection technique to inject Hu-MuSC salong with 50 µl 0.5% bupivacaine directly into the muscle of one leg. For cell injection, a 31 gauge needle on a 50 µl Hamilton syringe was used. Equal numbers of cells were injected into each experimental leg within experiments, but varied slightly between experiments as indicated in the text. The skin was closed with sutures and skin glue was applied over the incision. When multiple injections were utilized, Hu-MuSCs were suspended in 50 µl of in 0.5% bupivacaine and then subsequently transplanted in nine injections of approximately 5.5 µl per NSG TA. The transplant sites were spaced evenly apart in a grid of three by three injections, covering the majority of the TA muscle. Transplanted TA muscles were harvested at designated time points after transplantation. Harvested muscles were frozen in 2-methylbutane chilled in liquid nitrogen. Serial 6 µm transverse sections of the whole muscle were analyzed.

### Satellite cell sorting

Freshly harvested human muscle was either immediately digested or stored in DMEM with 30% FBS at 4˚C. Muscle was trimmed of excess fat, tendon, connective tissue, and fascia and mechanically minced. The tissue was then digested in 1 mg/ml collagenase XI (Sigma-Aldrich) in Dulbecco's Modified Eagle Medium (DMEM) with high glucose, 10% FBS and 1% Penicillin/Streptomycin at 37˚C for 70 min with intermittent manual needle trituration, performed slowly with an 18-gauge needle. Digests were washed with PBS and further digested with 0.25% trypsin at 37˚C for 12–15 min. Suspensions were passed through 40 µm nylon mesh, erythrocytes were lysed with ACK lysing buffer (ThermoFisher) for 5–7 min on ice, and washed with PBS. Magnetic column depletion of hematopoietic and endothelial cells was performed after cells were stained with anti-CD45 and anti-CD31 magnetic beads (Miltenyi Biotec). This step has the added benefit of removing small fiber fragments and facial tissue, which are a cause of high background on the flow cytometer. Unbound cells were washed and stained with anti-CD29-488 or 647 (eBioscience), anti-CD31-450 (eBioscience), anti-CD34-450 (eBbioscience), anti-CD45-450 (eBbioscience), anti-CD56-APC-vio-770 (Miltenyi Biotec), and anti-CXCR4-PE or APC (eBbioscience) (*Note* for the reisolation of Hu-MuSCs from transplanted mice: mouse muscle was processed as stated for human muscle, stained with the following antibodies: anti-human CD29-488 or 647 (Ebioscience), anti-human CD31-450 (eBbioscience), anti-human

CD45-450 (eBbioscience), anti-human CD56-APC-vio-770 (Miltenyi Biotec), anti-human CXCR4-PE or APC (eBioscience), anti-mouse CD31-450 (eBioscience), anti-mouse CD45-450 (eBbioscience), and anti-mouse Sca1-450 (eBbioscience)). Cells were washed and resuspended in flow cytometry buffer with 1:1000 sytox blue (Life Technologies). Flow cytometry antibodies listed in the Key Resources Table. Flow cytometry analysis and cell sorting were performed at the University of California San Francisco Flow Cytometry Core with the BD FACSAria2 operated using FACSDiva software. Viable cells were depleted for CD31, CD34, and CD45 expressing cells. Cells that remained after depletion were sorted for CXCR4+/CD29+/CD56+ and collected for further experimentation. We have previously published FMO controls for CD56 and CD29 use in Hu-MuSC isolation (*Xu et al., 2015*). Cells were sorted in 20% FBS in DMEM supplemented with 10 µM Rho-associated protein kinase inhibitor (ROCKi) (Y-27632 2HCl, Selleck Chemicals). See (*Garcia et al., 2017*; *Xu et al., 2015*) for details of the authors' prior muscle digestion and Hu-MuSC isolation protocol. Flow cytometry isolations were analyzed with FACSDiva and FlowJo software. Satellite cell subpopulations were analyzed and/or sorted with the following antibodies: anti-human CAV1-PE (abcam), ICAM1-PE (abcam), VCAM1 (eBioscience) and DLK1 (R and Dsytems).

## NSG TA analysis

All glass slides were removed from −80°C and warmed at room temperature for 10 min. For human DYSTROPHIN immunostaining, sections were fixed in 4% PFA for 10 min at room temperature and then washed in PBST (PBS with 0.1% Tween-20 (Sigma-Aldrich). The sections were blocked with 10% goat serum in PBS for 10 min at room temperature. The sections were then incubated overnight at room temperature with mouse monoclonal anti-human DYSTROPHIN (1:10 DSHB), human specificity of which was previously confirmed (*Xu et al., 2015*). The sections were then washed in PBST followed by 1 hr of incubation at room temperature with Alexa Fluor 594 goat anti-mouse IgG (1:500 Thermo) in 10% normal goat serum in PBS. Sections were mounted with VECTASHIELD mounting medium with DAPI (Vector Laboratories) and all samples were examined using a Leica upright microscope. Human-derived fibers (e.g. hDYSTROPHIN positive) were quantified by counting the number of positively stained fibers in the section with the most positive fibers after analyzing sections along the length of the muscle as has been previously reported (*Rozkalne et al., 2014*; *Xu et al., 2015*). For all other immunostainings, the slides were fixed in 4% PFA at room temperature for 10 min, washed in PBST, and then blocked with protein-free serum block (DAKO) and incubated at room temperature overnight with the following primary antibodies: mouse monoclonal IgG1 anti-PAX7 (1:10 DSHB), rabbit polyclonal anti-LAMININ (1:250 Sigma-Aldrich), mouse monoclonal IgG2b anti-human SPECTRIN (Leica Microsystems), mouse monoclonal IgG2b anti-human LAMIN A/C (Vector Laboratories), DLK1(1:50 abcam), ICAM1(1:50 Santa Cruz Biotechnology), CYCS (1:100 LifeSpan Bioscience), VCAM1 (1:75 abcam), CAV1 (1:500 abcam) and Sprouty1 (1:50). After PBST wash the following corresponding secondary antibodies were applied for 1 hr at room temperature: FITC donkey anti-mouse (1:500 Jackson Immunology), Cy3 goat anti-mouse (1:500 Jackson Immunology), Cy5 donkey anti-mouse (1:500 Jackson Immunology), Cy5 donkey anti-rabbit (1:300 Jackson Immunology), Alexa Fluor 488 goat anti-mouse IgG1 (1:500 Thermo), Alexa Fluor 594 goat anti-mouse IgG1 (1:500 Thermo), Alexa Fluor 488 goat anti-mouse IgG2b (1:500 Thermo), Alexa Fluor 594 goat anti-mouse IgG2b (1:500 Thermo). Sections were mounted with VECTASHIELD mounting medium with DAPI (Vector Laboratories) and all samples were examined using a Leica upright microscope.

## Code and data availability

Single cell gene expression data have been deposited and can be found here: https://doi.org/10.7272/Q65X273X (*Pomerantz and Barruet, 2020*). Detailed scripts for each analysis are in *Source code 1–5*.

## Acknowledgements

This work was supported by the CIRM New Faculty Physician Scientist Award RN3-06504 and NIH R01AR072638-03 to JHP, the UCSF PROF-PATH program via NIH R25MD006832 to SMG, UCSF Research Allocation Program for trainees to SL, and the Eli and Edythe Broad Center of Regeneration Medicine and Stem Cell Research Fellowship to AW. This work was also supported by NIH grants (R56AR060868, R01AR076252) to ASB. The authors would like to express their thanks for the

cooperation of Donor Network West and all of the organ and tissue donors and their families, for giving the gift of life and the gift of knowledge, by their generous donation.

## Additional information

### Funding

| Funder | Grant reference number | Author |
|---|---|---|
| California Institute for Regenerative Medicine | New Faculty Physician Scientist Award RN3-06504 | Jason H Pomerantz |
| National Institutes of Health | R01AR072638-03 | Jason H Pomerantz |
| University of California, San Francisco | UCSF PROF-PATH program via NIH R25MD006832 | Steven M Garcia |
| University of California, San Francisco | Research Allocation Program for trainees | Solomon Lee |
| Eli and Edythe Broad Foundation | Eli and Edythe Broad Center of Regeneration Medicine and Stem Cell Research Fellowship | Alvin Wong |
| National Institutes of Health | R56AR060868 | Andrew S Brack |
| National Institutes of Health | R01AR076252 | Andrew S Brack |

The funders had no role in study design, data collection and interpretation, or the decision to submit the work for publication.

### Author contributions

Emilie Barruet, Data curation, Software, Formal analysis, Validation, Investigation, Visualization, Writing - original draft, Writing - review and editing; Steven M Garcia, Conceptualization, Data curation, Formal analysis, Investigation, Visualization, Writing - original draft; Katharine Striedinger, Sun Xuefeng, Formal analysis, Investigation; Jake Wu, Solomon Lee, Alvin Wong, Stanley Tamaki, Investigation; Lauren Byrnes, Data curation, Software, Validation; Andrew S Brack, Investigation, Resources, Writing - review and editing; Jason H Pomerantz, Conceptualization, Resources, Data curation, Supervision, Funding acquisition, Project administration, Writing – original draft preparation, Writing – review and editing

### Author ORCIDs

Emilie Barruet  https://orcid.org/0000-0002-4593-024X
Steven M Garcia  https://orcid.org/0000-0002-7833-6677
Jason H Pomerantz  https://orcid.org/0000-0002-5107-1883

### Ethics

Human subjects: This study was conducted under the approval of the Institutional Review Board at The University of California San Francisco (UCSF). Written informed consent was obtained from all subjects.
Animal experimentation: All procedures were approved and performed in accordance with the UCSF Institutional Animal Care and Use Committee (Protocols #181101).

### Decision letter and Author response

Decision letter https://doi.org/10.7554/eLife.51576.sa1
Author response https://doi.org/10.7554/eLife.51576.sa2

## Additional files

### Supplementary files

- Source code 1. Multiple dataset analysis.
- Source code 2. Pseudotime analysis.
- Source code 3. HEY1$^{high}$SPRY1$^{high}$ subsetting.
- Source code 4. CAV1$^{high}$ subsetting.
- Source code 5. Gene Ontology and Pathaway analyses.
- Supplementary file 1. Genes differentially expressed in each cluster for the combined vasti lateralis samples.
- Supplementary file 2. Genes differentially expressed in each cluster for the rectus femoris sample.
- Supplementary file 3. Genes differentially expressed in the *SPRY1/HEY1* high expressing satellite cells.
- Supplementary file 4. Genes differentially expressed in the *CAV1* high expressing satellite cells.
- Supplementary file 5. Type of muscle used per experiment.
- Transparent reporting form

### Data availability

Single cell RNA sequencing data were uploaded to Dryad and can be accessed here https://doi.org/10.7272/Q65X273X.

The following dataset was generated:

| Author(s) | Year | Dataset title | Dataset URL | Database and Identifier |
|---|---|---|---|---|
| Barruet E, Garcia SM, Striedinger K, Wu J, Lee S, Byrnes L, Wong A, Xuefeng S, Tamaki S, Brack AS, Pomerantz JH | 2020 | Functionally heterogeneous human satellite cells identified by single cell RNA sequencing | https://doi.org/10.7272/Q65X273X | Dryad Digital Repository, 10.7272/Q65X273X |

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
