## [Decision Letter]

**Acceptance summary:**

The authors have done an impressive work in revising the manuscript substantially, adding a substantial number of human samples that allowed a more comprehensive analysis of human muscle satellite cells. Of note, human satellite cell subpopulations are described that are Cav1+ and Cav1-, and transplantations of these cells in mice showed a greater engraftment potential of the Cav1+ subclass. In summary, these findings provide a valuable resource for the community.

**Decision letter after peer review:**

[Editors’ note: the authors submitted for reconsideration following the decision after peer review. What follows is the decision letter after the first round of review.]

Thank you for submitting your work entitled "Functionally heterogeneous human satellite cells identified by single cell RNA sequencing" for consideration by *eLife*. Your article has been reviewed by three peer reviewers, one of whom is a member of our Board of Reviewing Editors, and the evaluation has been overseen by a Reviewing Editor and a Senior Editor. The following individuals involved in review of your submission have agreed to reveal their identity: Bradley B Olwin (Reviewer #3).

Our decision has been reached after consultation between the reviewers. Based on these discussions and the individual reviews, we regret to inform you that your work will not be considered further for publication in *eLife*.

In this study, the authors combine single cell RNA sequencing and flow cytometry to analyze human satellite cells isolated from skeletal muscles. A single sample from a middle-aged adult (56y old) and from an aged adult (86y old) were subjected to single cell sequencing. They report that the SCs contains transcriptionally distinct subpopulations. Using pseudotime analysis, they show that myogenic cells can be ordered into various state, from quiescent stem cells to more differentiated progenitor cells. Genes such as DLK1, ICAM1, CYCS and VCAM1 are differentially expressed in human SCs and VCAM1 expression is increased in human SC isolated from aged muscle. They also identify Caveolin1 (CAV1) as a cell surface marker to sort the more quiescent human SCs (CAV1+) and following their transplantation in immunodeficient mice, report a better regeneration compared to CAV- cells.

Therefore, this study identifies some differentially expressed markers and properties that bring some new knowledge to the field. However, some of the analysis and interpretation of the results needs attention (see below). The reviewers understand the challenges involved in analyzing human samples, however, given the small sample size and the fact that only a portion of a muscle is taken for analysis, it is unclear how widely applicable these observations will be. The reviewers feel that the amount of work that will be required to verify these findings will not reasonably fit within the context of a revision. Therefore, publication of the study in its current form in *eLife* is not recommended.

Essential revisions:

1) The study comes from analysis of one individual (1 – 84-year old, 1 – 56-year old), therefore, 84-year old specific clusters should not be generalized to aging muscles. The authors are also comparing 2 different muscles between adult and aged (rectus femoris and vastus lateralis) which impact the interpretation of the results due to known inter-muscle heterogeneity.

2) In 2 recent papers (Vartanian et al., 2019; Scaramozza et al., 2019), Pax3 was shown to be expressed and enriched in a minor subset of murine Pax7+ SC, conferring functional heterogeneity in SC population. Have the authors evaluated Pax3 expression in the distinct subpopulation of human SCs? Such data could be added in Figure 1 or at least mentioned in the Discussion section.

3) It was shown (Machado et al., 2017) that quiescent muscle stem cells undergo major transcriptomic alterations during the isolation process, enough to induce biochemical changes. The use of the term "quiescent" throughout the paper should be qualified, since the authors do not address the issue of quiescence of freshly isolated human SC, or show that SCs are in G0.

4) Concerns regarding heterogeneity: one might expect heterogeneity in the SC population as some would respond to exercise or injury and some part of the population would be quiescent. None of the data provided disproves that what the authors observe is simply a continuum of SC behavior and the heterogeneity is a result of cells in continuous flux. Biological replicates, perhaps obtained from the same individual and different muscle groups, would help to address this issue. The data as presented in the manuscript imply that separable and heterogeneous SC pools are present, while a counter argument is that this is simply a continuum in constant flux.

5) Also, are SCs present that are not isolated as their relative expression of CD56 and CD29 are low? Does single cell sequencing of the entire mononuclear cell population from muscle corroborate the heterogeneity data presented? How does flow cytometry affect gene expression in SCs? It is possible and even likely that the heterogeneity observed could in part be derived from the isolation and sorting of SCs.

6) The scale is lacking in all the immuno-fluorescent pictures shown in Figure 2, Figure 3, Figure 4 and Figure 5 and/or in the figure legends.

7) The method used to merge the data might be problematic: normally, when data come from the same 10x chip and from the same sequencing lane (which is the case in the experiment) the Seurat MergeSeurat function is sufficient. However, in Figure 3E, there is a clear separation by individual. Specifically, subsection “VCAM1 is differentially expressed on satellite cells of aged muscle in single cell transcriptomes and in vivo*”*, there is mention of a batch effect correction without mentioning which one was used. Authors should also try the MNN (Mutual Nearest Neighbors) and/or the CCA (Canonical Correlation Analysis) algorithms to see if these could help in correcting the batch effect.

8) Cluster 4 in the 84 year-old individual looks like it contains a little bit of everything, which can fit what we know about evolution of transcriptome regulation during ageing. But it can also arise from bad quality barcodes i.e, no cell, specially knowing that the authors chose to set a very low number (200) of expressed genes in their analysis, these can also correspond to barcodes with too many genes expressed (information about this cutoff is missing) which can correspond to doublets.

9) Figure 1D and Figure 3C: According to the size of the dots on the Dotplot (showing normalized proportions, and not% of Expression as indicated), only 20-40% of satellite cells seem to express Pax7. The authors should comment on this point to place the work in the context of the mouse and could provide a tSNE plot of Pax7 expression across all 5062 cells. Is this due to a possible lack of sensitivity in the sequencing?

10) Figure 1G and H: Pseudotime is used to compute artificially the progression of a lineage through differentiation (during embryonic development or adult stem cells). Monocle 2 analysis here brings confusion to the results: cells belonging to the "satellite cells" clusters appear on the same "branch" as mesenchymal cells and have a lower pseudotime, as if they were progenitors of these cells. By representing their data in this fashion, the authors imply that these cells belong to the same lineage in human resting muscle (satellite cells differentiating into mesenchymal cells). If the authors want to show progress through myogenesis, they need to perform this analysis on myogenic cells only, excluding fibroblastic cells (clusters 0,1,2,3,4,6).

11) The authors claim a progression through myogenesis from cluster 0,1,2 to 4 and 6. However the t-SNE plot shows a very nebular distribution of these populations, especially a closer transcriptomic proximity of clusters 1, 4 and 6 as opposed to 3. Removing fibroblastic cells in this representation could allow better highlight of intra-myogenic transcriptomic diversity and similarity.

12) Figure 1C is hard to read (and not convincing). Combining Figure 1D and E would be more informative.

13) Figure 3D: How was this correlation performed? The correlations of cluster 4 of the aged are quite similar to the correlations found in clusters 0, 1, 2 and 3.

14) Figure 3E: The merged data shows multiple clusters primarily made of either Aged (clusters 4,6) or Adult (2,3 and 5) cells. How do the authors explain such differences when correlations shown in Figure 3D seem so high? Why did the authors focus on cluster 6 specifically when numerous clusters do not match? Displaying the proportion of cell origin for each cluster would be informative here to assess this mismatch.

15) In Figure 3G, please provide a better image for Pax7/VCAM1 expression to support the conclusion that VCAM1 is express more frequently in SC of aged muscle (images at lower magnification).

16) Can the authors provide measurements of UMI counts, gene counts and cycling score for each cluster? These variables are often found to influence clustering analysis and did not seem to have been regressed out during scaling of the data, judging by the Material and methods section.

17) Figure 4B violin plots seem to suggest a high expression of Myod1 and Myf6 in the Hey1+/Spry1+ population which is the opposite of what the authors claim.

18) Given that the isolation strategy the authors used also captures mesenchymal cells, Cav1 may be expressed preferentially in myogenic cells, thus enriching the myogenic yield of the isolation approach, independently of a more "quiescent" state of satellite cells. The authors need to show that Cav1 does not preferentially select the myogenic compartment.

19) The point concerning the robust engraftment of CAV1+ human SC should be extensively discussed in regard to the numerous papers describing human myogenic stem cell engraftment after in vivo implantation in immunodeficient mice. Could you also clarify if injected human SC are isolated from the same donor? By flow cytometry, the CAV1+ SC represent 51.6% of the CD29/CD56 population (Figure 5). It would be interesting to know the percentage of the CAV1+ SC related to the live cell population (FSC/SSC gated population) obtained after muscle dissociation.

20) Also, regarding the CAV1+ population, in Figure 4D, this population appears to be 80% of the SCs. When sorted, the percentage drops, which is not surprising due to the harsh conditions encountered when sorting cells. Thus, this population simply represents most SCs with a subset exhibiting poor engraftment. There have been a number of publications demonstrating that good engraftment can be achieved even with low numbers of SCs by sorting for specific markers, by transplanting intact myofibers, by transplanting SCs in engineered gels, or by the use of specific inhibitors to maintain SCs in quiescence upon isolation. They should refer to Arpke and Kyba, 2016 and 2012 which demonstrate that small numbers of cells are effective for transplantation.

21) The images provided in Figure 5E where few of the human spectrin lamin a/c+ cells appear as SCs, the majority appear interstitial in the provided image. Few are Pax7+ and thus, it is difficult to determine how the quantification was performed. Insufficient experimental detail is provided to assess which cells were transplanted and how the cells were derived. Are the biological replicates referred to in the figure from 3 different human individuals or are these 3 samples from one individual? If from one individual, then these are not biological replicates but technical transplantation replicates. The figure title states transplantation is robust and the data show the numbers of transplanted myofibers that are dystrophin+. However, if the data were plotted as a percentage of the total myofiber number in the TA muscle it is unclear how robust the transplantation is as 75 dys+ myofibers/~3500 myofibers per TA is ~2% of the total. If plotted as a percentage of the total myofibers per TA muscle or as a total of the SC number per myofiber are the data sufficient to establish that they are significantly different between the samples?

---

## [Author Response]

[Editors’ note: The authors appealed the original decision. What follows is the authors’ response to the first round of review.]

Essential revisions:1) The study comes from analysis of one individual (1 – 84-year old, 1 – 56-year old), therefore, 84-year old specific clusters should not be generalized to aging muscles. The authors are also comparing 2 different muscles between adult and aged (rectus femoris and vastus lateralis) which impact the interpretation of the results due to known inter-muscle heterogeneity.

The reviewer raises an important point which is relevant to all publications using single cell sequencing of muscle to date: How many samples are required and is it valid to pool muscle groups. We now provide scRNA-seq profiles from 8 different vastus lateralis muscles from 8 individuals (age range (20-83), addressing reasonable questions of heterogeneity related to individuals. Data from these samples is used in new Figure 1, Figure 2, Figure 3, Figure 4 and Figure 5A. We also provide duplicates from rectus abdominis and pectoralis major muscles, thus addressing the issue of muscle group heterogeneity (new Figure 2A, Figure 5A, Figure 1—figure supplement 1B, Figure 2—figure supplement 1). Due to the concern of sample variation we have removed the single rectus femoris sample from the primary analysis and include that data in a separate figure (Figure 1—figure supplement 2C, Figure 2—figure supplement 1C and Figure 3—figure supplement 1). Our analysis of each of these 13 muscles, all from different individuals, found similar transcriptional clusters, including the CAV1 clusters that we report to be functionally distinct. Thus, we are able to provide strong data, confirming that our findings can be generalized to adult human muscles. This level of single cell transcriptomic analysis has not been performed on any species to date.

This issue of aging was also raised as a possible source of variation, to this end we have removed conclusions dependent on limited samples from aged individuals. Although our analysis of VCAM expression in multiple human samples supports conclusions for that protein, we agree with the reviewer’s comment that the number of samples that would be required to quantitatively show generalized differences in transcriptional clustering is prohibitively high, and we therefore removed conclusions about transcriptional changes with aging from the manuscript.

2) In 2 recent papers (Vartanian et al., 2019; Scaramozza et al., 2019), Pax3 was shown to be expressed and enriched in a minor subset of murine Pax7+ SC, conferring functional heterogeneity in SC population. Have the authors evaluated Pax3 expression in the distinct subpopulation of human SCs? Such data could be added in Figure 1 or at least mentioned in the Discussion section.

Based on the reviewer’s query, we analyzed Pax3 expression. We find detectable Pax3 expression in all the SC clusters of the myogenic compartment. We have included this in the text of the Results section and in Figure 1—figure supplement 2.

3) It was shown (Machado et al., 2017) that quiescent muscle stem cells undergo major transcriptomic alterations during the isolation process, enough to induce biochemical changes. The use of the term "quiescent" throughout the paper should be qualified, since the authors do not address the issue of quiescence of freshly isolated human SC, or show that SCs are in G0.

We thank the reviewer for raising this important issue. On reflection the data from human muscle should be interpreted with more caution than mice in a controlled environment. Therefore, we no longer refer to these cells as quiescent, but simply as human satellite cells in the Abstract and throughout the manuscript text.

We also acknowledge the importance of clearly stating that transcriptomic alterations are likely to happen with human satellite cell isolation, and this is discussed in the Discussion section. In the revised manuscript we state that the study relates to transcriptional profiling of freshly isolated, uncultured human satellite cells. Throughout the manuscript careful attention is now given to articulating precisely where quiescence is assessed. We also include new data that shows cell cycle profiling (Figure 3—figure supplement 1) of isolated cells. It is also important to relate the transcriptomic state of human samples to known markers from murine SCs. To this end, we analyzed known G0 cell cycle and quiescence markers (Figure 3B and C). We also show that there is a small but distinct population (subpopulation 15, Figure 1) of activated satellite cells in our samples which serve as a reference for characterization of the main populations in terms of degree of relative quiescence.

Furthermore, we provide several additional functional assays for proliferation in vitro showing that Ki67 is expressed significantly less in CAV1+ SCs, MYOD expression is lower and that time to first division is longer in CAV1+ SCs. These data are shown in new (Figure 5G-K). Finally, we have provided further analysis of SPROUTY1. We now show that SPROUTY1 protein is expressed in a large fraction of (but not all) Pax7+ cells in fixed muscle sections (Figure 3D).

4) Concerns regarding heterogeneity: one might expect heterogeneity in the SC population as some would respond to exercise or injury and some part of the population would be quiescent. None of the data provided disproves that what the authors observe is simply a continuum of SC behavior and the heterogeneity is a result of cells in continuous flux. Biological replicates, perhaps obtained from the same individual and different muscle groups, would help to address this issue. The data as presented in the manuscript imply that separable and heterogeneous SC pools are present, while a counter argument is that this is simply a continuum in constant flux.

We acknowledge this consideration of the reviewer. In the revised manuscript we now show transcriptional clusters across 13 samples. Cav1 is expressed in a subset in each sample. Therefore, the population is heterogeneous and, as we show by flow cytometry, physically separable. However, this does not exclude a model whereby SCs exist in a continuum of states. Differences in transcript/translational kinetics would regulate the stability of a behavioral state. While we find both possibilities fascinating, they cannot be parsed using a static and destructive method such as single cell sequencing. This is discussed in depth in the revised Discussion section.

5) Also, are SCs present that are not isolated as their relative expression of CD56 and CD29 are low? Does single cell sequencing of the entire mononuclear cell population from muscle corroborate the heterogeneity data presented? How does flow cytometry affect gene expression in SCs? It is possible and even likely that the heterogeneity observed could in part be derived from the isolation and sorting of SCs.

Although we have previously published several lines of evidence that in human muscles, all or the vast majority of satellite cell characteristics and function reside within the isolated CD56 and CD29 compartment (Xu, 2015 Stem Cell Reports), we cannot exclude that some of the heterogeneity observed may be due to the isolation and sorting. Our approach of enzymatic digestion and flow cytometry is a conventional approach in the field to isolate satellite cells. We have not analyzed entire mononuclear fraction from human muscle. As discussed above the isolation of cells does invoke activation. Unfortunately, single cell sequencing analysis using 10x does not work on fixed cells or on tissues. This is a technical limitation not restricted to this manuscript. However, we do find heterogeneous Cav1 and SPROUTY1 at protein level on sections (Figure 3B and Figure 5B), which supports our conclusions from scRNA-seq.

The data in Figure 2 and Figure 3 which validate the transcriptional clusters by using immunostaining of muscle sections to confirm in vivo the observed transcriptional heterogeneity, address this question. We show data for different markers representing different clusters that clearly support the observed transcriptional heterogeneity as a property of satellite cells in vivo prior to isolation. The data show that some SCs respond differently to functional assays including transplantation, further supporting that we have subtypes.

Additionally, we now present new data of in vitro analysis of CAV1+ vs CAV1- time to first division, and Ki67 staining (Figure 5), which add highly substantial information in support of functional heterogeneity.

6) The scale is lacking in all the immuno-fluorescent pictures shown in Figure 2, Figure 3, Figure 4 and Figure 5 and/or in the figure legends.

Thank you for pointing this out, all our images now have appropriate scale bars which are defined in the figure legends.

7) The method used to merge the data might be problematic: normally, when data come from the same 10x chip and from the same sequencing lane (which is the case in the experiment) the Seurat MergeSeurat function is sufficient. However, in Figure 3E, there is a clear separation by individual. Specifically, subsection “VCAM1 is differentially expressed on satellite cells of aged muscle in single cell transcriptomes and in vivo”, there is mention of a batch effect correction without mentioning which one was used. Authors should also try the MNN (Mutual Nearest Neighbors) and/or the CCA (Canonical Correlation Analysis) algorithms to see if these could help in correcting the batch effect.

With our new set of samples, we performed analyses using Seurat v3.1.2 which now have a new non-linear batch correction algorithm. Additional information can be found in the Single Cell RNA Sequencing and Analysis section of the Materials and methods section.

Please name the function used and the arguments in the function (default or specifically selected). It is advisable to provide access to scripts for future reproducibility.

Functions and arguments in the function are named in the subsection “Single Cell RNA Sequencing and Analysis”. In addition, we provide all scrips used for our analyses.

8) Cluster 4 in the 84 year-old individual looks like it contains a little bit of everything, which can fit what we know about evolution of transcriptome regulation during ageing. But it can also arise from bad quality barcodes i.e, no cell, specially knowing that the authors chose to set a very low number (200) of expressed genes in their analysis, these can also correspond to barcodes with too many genes expressed (information about this cutoff is missing) which can correspond to doublets.

We now include 12 new samples. We increased our number of expressed genes to 500 in all our sample analyses to exclude poor quality barcodes as a source of variability. We also excluded cells expressing more than 6000 genes to account for potential doublets (Materials and methods section). Additionally, clusters expressing feature that clearly defined two different cell type were excluded from downstream analysis (cluster 11). We provide additional information on used parameters in the Materials and methods section.

9) Figure 1D and Figure 3C: According to the size of the dots on the Dotplot (showing normalized proportions, and not% of Expression as indicated), only 20-40% of satellite cells seem to express Pax7. The authors should comment on this point to place the work in the context of the mouse and could provide a tSNE plot of Pax7 expression across all 5062 cells. Is this due to a possible lack of sensitivity in the sequencing?

We have previously published that satellite cells isolated according to the protocol used in this study uniformly express PAX7 by immunostaining (Garcia et al., 2018). Therefore, Pax7 transcript likely reflects limitations of single cell sequencing sensitivity. Indeed, similar observations were made in a recent publication on mouse satellite cells (Dell’Orso et al., 2019). In the text of the revised manuscript we discuss this, and show that with the newer kit used for the revised analyses, PAX7 transcript is detected in a much higher proportion of satellite cells (Figure 6 legend). Finally, we added a feature plot depicting PAX7 expression in the eight vasti (Figure 1—figure supplement 2A).

10) Figure 1G and H: Pseudotime is used to compute artificially the progression of a lineage through differentiation (during embryonic development or adult stem cells). Monocle 2 analysis here brings confusion to the results: cells belonging to the "satellite cells" clusters appear on the same "branch" as mesenchymal cells and have a lower pseudotime, as if they were progenitors of these cells. By representing their data in this fashion, the authors imply that these cells belong to the same lineage in human resting muscle (satellite cells differentiating into mesenchymal cells). If the authors want to show progress through myogenesis, they need to perform this analysis on myogenic cells only, excluding fibroblastic cells (clusters 0,1,2,3,4,6).

We thank the reviewer for this suggestion. We provide the new analysis on myogenic cells only in the revised manuscript (new Figure 3 and Figure 3—figure supplement 1D,E).

11) The authors claim a progression through myogenesis from cluster 0,1,2 to 4 and 6. However the t-SNE plot shows a very nebular distribution of these populations, especially a closer transcriptomic proximity of clusters 1, 4 and 6 as opposed to 3. Removing fibroblastic cells in this representation could allow better highlight of intra-myogenic transcriptomic diversity and similarity.

Our new analysis of multiple samples along with removal of non-myogenic cells does indeed better highlight intra-myogenic diversity and similarity. This analysis is included in the revised manuscript (Figure 3E,F and Figure 3—figure supplement 1D,E). The UMAP and the pseudotime analysis provide a better representation of the clusters containing cells at different stages of myogenic progression from stem cell to differentiated muscle cell.

12) Figure 1C is hard to read (and not convincing). Combining Figure 1D and E would be more informative.

Thank you for this suggestion, we removed Figure 1C.

13) Figure 3D: How was this correlation performed? The correlations of cluster 4 of the aged are quite similar to the correlations found in clusters 0, 1, 2 and 3.

The new analysis on multiple samples does not require correlation analysis as we are not making conclusions with respect to aging and therefore it has been removed.

14) Figure 3E: The merged data shows multiple clusters primarily made of either Aged (clusters 4,6) or Adult (2,3 and 5) cells. How do the authors explain such differences when correlations shown in Figure 3D seem so high? Why did the authors focus on cluster 6 specifically when numerous clusters do not match? Displaying the proportion of cell origin for each cluster would be informative here to assess this mismatch.

We now provide data for 13 samples as well as the distribution of cells of each cluster per sample (Figure 1B,C). The new distribution data across this large number of samples clearly demonstrates consistency of major clusters across individuals. As discussed, we removed conclusions with respect to aging from the manuscript.

15) In Figure 3G, please provide a better image for Pax7/VCAM1 expression to support the conclusion that VCAM1 is express more frequently in SC of aged muscle (images at lower magnification).

Due to the lack of replicates for transcriptome analyses, all data assessing VCAM expression during aging has been removed from the manuscript.

16) Can the authors provide measurements of UMI counts, gene counts and cycling score for each cluster? These variables are often found to influence clustering analysis and did not seem to have been regressed out during scaling of the data, judging by the Material and methods section.

We added the UMI counts (RNA count) and cycling score for each myogenic cluster (Figure 3—figure supplement 1B,C). We regressed out heterogeneity associated with mitochondrial contamination and UMI counts but not with cell cycle stage since in satellite cells assessing cell cycle markers is important to characterize quiescence/activation and therefore clusters. Gene count (or nFeature_RNA) is included below for the reviewers, but are not included in the manuscript since we stated our filtering process in the Materials and methods section. We can include this information in the manuscript if the reviewers feel that this information would be useful.

**Author response image 1. respfig1:** Gene count per cluster.

17) Figure 4B violin plots seem to suggest a high expression of Myod1 and Myf6 in the Hey1+/Spry1+ population which is the opposite of what the authors claim.

We have repeated this analysis with the extra samples and confirm that Myod1 has lower expression in the Hey1/Spry1 population (Figure 3H), which supports our conclusions regarding this population.

We had originally used in silico analysis to that conclude Cav1 expression was elevated in Hey1hi/Spry1hi populations. After further analysis on 8 samples the results changed somewhat. We no longer find increased expression levels but instead a greater variable distribution of Hey1^hi^ cells expressing Cav1 and Spry1. This is supported by immunohistochemistry on fixed tissues that shows that a fraction of human satellite cells expressed Spry1 (Figure 3D) and Cav1 (Figure 5B). This manuscript is focused on the functional heterogeneity in human SCs, therefore these new data do not impact the conclusions of our findings.

18) Given that the isolation strategy the authors used also captures mesenchymal cells, Cav1 may be expressed preferentially in myogenic cells, thus enriching the myogenic yield of the isolation approach, independently of a more "quiescent" state of satellite cells. The authors need to show that Cav1 does not preferentially select the myogenic compartment.

The number of contaminating mesenchymal cells that are captured by the isolation strategy is very small, representing less than 2% of the isolated cells, as confirmed by our new analysis of multiple samples, so there is no related enrichment of the myogenic yield with the isolation approach. We find that Cav1+ and Cav1- SCs (selected by Cd56+/CD29) both express Pax7 protein (Figure 5—figure supplement 1), clearly indicating that Cav1 does not select for a myogenic compartment. Moreover, GO analysis reveals that Cav1- SCs are enriched for categories involved in myogenesis (Figure 4D).

19) The point concerning the robust engraftment of CAV1+ human SC should be extensively discussed in regard to the numerous papers describing human myogenic stem cell engraftment after in vivo implantation in immunodeficient mice. Could you also clarify if injected human SC are isolated from the same donor? By flow cytometry, the CAV1+ SC represent 51.6% of the CD29/CD56 population (Figure 5). It would be interesting to know the percentage of the CAV1+ SC related to the live cell population (FSC/SSC gated population) obtained after muscle dissociation.

We have referenced our transplant data with those of published human data. We address this point with a table recapitulating the muscle type and donor used for each experiment (Supplementary file 5). The percentage of the CAV1+ SC related to the live cell population after muscle dissociation was 0.7 ± 0.3% (Discussion section). The Discussion section includes mention of description that places the new data into context with our prior studies using comparable experimental approaches, very small numbers of human satellite cells, and in relation to prior work by other groups including references. Comparisons of CAV1+ and CAV1- satellite cells are from the same donor in each individual experiment. At least 3 separate experiments with different donors were performed. This is clarified in the text in (Figure 5 and Figure 6 legends).

20) Also, regarding the Cav1+ population, in Figure 4D, this population appears to be 80% of the SCs. When sorted, the percentage drops, which is not surprising due to the harsh conditions encountered when sorting cells. Thus, this population simply represents most SCs with a subset exhibiting poor engraftment. There have been a number of publications demonstrating that good engraftment can be achieved even with low numbers of SCs by sorting for specific markers, by transplanting intact myofibers, by transplanting SCs in engineered gels, or by the use of specific inhibitors to maintain SCs in quiescence upon isolation. They should refer to Arpke and Kyba, 2016 and2012 which demonstrate that small numbers of cells are effective for transplantation.

We now provide new data from more samples that shows that the CAV1 population ranges in different samples (Discussion section). These cells exhibit high engraftment on a per cell basis, relative to CAV1-. Although our study deals with human satellite cells, in response to the reviewer’s comment we have included a discussion of mouse satellite cell transplants including the references mentioned above (Discussion section).

The reviewer raises a good point that we did not fully explain in methods. The sort is based on a cell surface stain, while IF with Pax7 is on permeabilized tissue. Therefore, the discrepancy between the two methods may reflect the fraction of cells that express intracellular Cav1 or downregulation of CAV1 during isolation (Discussion section).

21) The images provided in Figure 5E where few of the human spectrin lamin a/c+ cells appear as SCs, the majority appear interstitial in the provided image. Few are Pax7+ and thus, it is difficult to determine how the quantification was performed. Insufficient experimental detail is provided to assess which cells were transplanted and how the cells were derived. Are the biological replicates referred to in the figure from 3 different human individuals or are these 3 samples from one individual? If from one individual, then these are not biological replicates but technical transplantation replicates. The figure title states transplantation is robust and the data show the numbers of transplanted myofibers that are dystrophin+. However, if the data were plotted as a percentage of the total myofiber number in the TA muscle it is unclear how robust the transplantation is as 75 dys+ myofibers/~3500 myofibers per TA is ~2% of the total. If plotted as a percentage of the total myofibers per TA muscle or as a total of the SC number per myofiber are the data sufficient to establish that they are significantly different between the samples?

To quantify satellite cells, we used standard methodology in the field to identify and count sublaminar, PAX7+ mononucleated cells. The data were obtained using the same methodology we and others have published previously (Xu et al., 2015, Garcia et al., 2018) (Introduction), which is referenced. To clarify our methods, we added complete methodological description in the subsection “Supplemental Experimental Procedures”. Also, as is standard in the field, the data show replicates from a single experiment (different mice transplanted with CAV1+ and CAV1- satellite cells from a single human donor). The entire experiment was performed three separate times (three donors), which we clarify in the text (Figure 6 legend) addresses the question of biological replicates. The data presentation is standard to compare engraftment among experimental groups and we propose to reference prior publications and to more clearly explain the approach in the text.